



**Carbon dynamics in the Mekong Delta**

Alberto V. Borges[1], Gwenaël Abril[2,3], Steven Bouillon[4]

[1] Chemical Oceanography Unit, University of Liège, 4000 Liège, Belgium
[2] Programa de Geoquímica, Universidade Federal Fluminense, 24020015, Niterói, Brazil
[3] Laboratoire Environnements et Paléoenvironnements Océaniques et Continentaux, CNRS, Université de Bordeaux, 33405, Talence, France.
[4] Department of Earth and Environmental Sciences, KU Leuven, 3001 Leuven, Belgium

Correspondence to: Alberto V. Borges (alberto.borges@ulg.ac.be)



## Abstract

We report a data-set obtained in the three branches (My Tho, Ham Luong, Co Chien) of the Mekong delta (Bến Tre province, Vietnam) in December 2003, April 2004, and October 2004, of biogeochemical variables related to carbon cycling (pH, total
alkalinity (TA), $O_2$ saturation level (%$O_2$), calculated partial pressure of $CO_2$ (p$CO_2$), dissolved $CH_4$ concentration, particulate (POC) and dissolved (DOC) organic carbon concentration and stable isotope composition ($\delta^{13}$C-POC, $\delta^{13}$C-DOC), particulate nitrogen (PN), dissolved inorganic carbon (DIC) stable isotope composition ($\delta^{13}$C-DIC), total suspended matter (TSM)). Both the inner estuary (upstream of the mouth)
and the outer estuary (river plume) were sampled, as well as side channels. The values of p$CO_2$ ranged between 232 and 4,085 ppm, %$O_2$ between 63 and 114 %, and $CH_4$ between 2 and 2,217 nmol $L^{-1}$, within the ranges of values previously reported in temperate and tropical macro-tidal estuaries. Strong seasonal variations were observed. In the upper oligohaline estuary, low p$CO_2$ (479-753 ppm) and high
15   %$O_2$ (98-106%) values were observed in April 2004 most probably related to freshwater phytoplankton growth owing to low freshwater discharge (1,400 $m^3$ $s^{-1}$) and increase of water residence time; during the two other sampling periods with a higher freshwater discharge (9,300-17,900 $m^3$ $s^{-1}$), higher p$CO_2$ (1,895-2,664 ppm) and lower %$O_2$ (69-84%) values were observed in the oligohaline part of the estuary.
During the October 2004 sampling, important phytoplankton growth occurred in the off-shore part of the river plume as attested by changes in the contribution of POC to TSM (%POC), $\delta^{13}$C-POC, POC:PN ratios, possibly related to low TSM values (improvement of light conditions for phytoplankton development), leading to low p$CO_2$ (232 ppm) and high %$O_2$ (114%) values. Water in the side channels in the Mekong
delta was strongly impacted by inputs from the extensive shrimp farming ponds. The values of p$CO_2$, $CH_4$, %$O_2$, $\delta^{13}$C-DIC indicated intense organic matter degradation that was partly mediated by sulfate reduction (presumably in sediments), as indicated by the slope of TA and DIC co-variations. The $\delta^{13}$C-POC variations also indicated intense phytoplankton growth in the side channels, presumably due to nutrient
enrichment related to the shrimp farming ponds. A dataset in the mangrove creeks of the Ca Mau province (part of the Mekong delta) was also acquired in April 2004 and October 2004. These data extended the range of variability of p$CO_2$ and %$O_2$ with more extreme values than in the Mekong delta (Bến Tre), with maxima and minima of





6,912 ppm and 37%, respectively. Similarly, the maximum $CH_4$ concentration (686 nmol $L^{-1}$) was higher in the Ca Mau province mangrove creeks than in the Mekong delta (Bến Tre, maximum 222 nmol $L^{-1}$), during the October 2004 cruise (rainy season and high freshwater discharge period). In April 2004 (dry season and low freshwater discharge period), the $CH_4$ values were much lower than in October 2004 (average 19±13 and 210±158 nmol $L^{-1}$, respectively) in the Ca Mau province mangrove creeks, owing to the higher salinity (average 33.2±0.6 and 14.1±1.2, respectively) that probably led to higher sediment sulfate reduction, leading to inhibition of sediment methanogenesis and higher anaerobic $CH_4$ oxidation. In the inner estuarine region (three branches of the Mekong delta), $CO_2$ emissions to the atmosphere averaged 121 mmol $m^{-2}$ $d^{-1}$, and the $CH_4$ emissions averaged 118 µmol $m^{-2}$ $d^{-1}$. The $CO_2$ emission to the atmosphere from the Mekong inner estuary was higher than reported in the Yangtze and Pearl River inner estuaries. This was probably due to the lower salinity in the Mekong delta branches, possibly due to different morphology; relatively linear channels in the Mekong delta versus funnel-shaped estuaries for the Yangtze and Pearl River inner estuaries.



1. **Introduction**

Estuaries are the main pathways for the transfer of particulate and dissolved matter between land to the ocean (through rivers). Particulate and dissolved matter undergo strong transformations, as estuaries are sites of intense biogeochemical processing (for example, Bianchi, 2006) that in most cases leads to substantial emissions of greenhouse gases such as carbon dioxide ($CO_2$) and methane ($CH_4$) (for example, Borges and Abril, 2011). Most estuarine environments are net heterotrophic ecosystems (for example, Gattuso et al., 1998), leading to the production and emission to the atmosphere of $CO_2$ and $CH_4$. The production of $CO_2$ and $CH_4$ is modulated by various physical features resulting from estuarine geomorphology such as water residence time (Borges et al., 2006; Joesoef et al., 2017), tidal amplitude and vertical stratification (Borges, 2005; Koné et al., 2009; Crosswell et al., 2012; Joesoef et al., 2015), and connectivity with tidal flats and saltmarshes (Middelburg et al., 2002; Cai, 2011). Highly eutrophic (Cotovicz Jr et al., 2015) or strongly stratified estuarine systems (Koné et al., 2009) can exceptionally act as sinks of $CO_2$ due to high carbon sequestration, although high organic matter sedimentation can concomitantly lead to high $CH_4$ production and emission to the atmosphere (Koné et al., 2010; Borges and Abril, 2011).

The global $CO_2$ emissions from estuaries have been estimated by several studies (Abril and Borges, 2004; Borges 2005; Borges et al., 2005; Chen and Borges, 2009; Laruelle et al., 2010; 2013; Cai, 2011; Chen et al., 2012; 2013) and range from 0.1 to 0.6 PgC yr$^{-1}$, equivalent in magnitude to 5-30% of the oceanic $CO_2$ sink of ~2 PgC yr$^{-1}$ (Le Quéré et al., 2016). These values were derived from the scaling of air-water $CO_2$ flux intensities (per surface area) compiled from published data that were extrapolated to estimates of the global surface of estuaries. The most recent estimates are lower than the older ones, reflecting the increase by an order of magnitude of the availability of data on air-water $CO_2$ fluxes, and more precise estimates of surface areas of estuaries structured by typology (for example, Dürr et al., 2011). The global estimates of $CH_4$ emissions from estuaries are also relatively variable ranging between 1 and 7 TgCH$_4$ yr$^{-1}$ (Bange et al., 1994; Upstill-Goddard et al., 2000; Middelburg et al., 2002; Borges and Abril, 2011), and are modest compared to other natural (220-350 TgCH$_4$ yr$^{-1}$) and anthropogenic (330-335 TgCH$_4$



yr$^{-1}$) CH$_4$ emissions (Kirschke et al., 2013). Unlike CO$_2$, the most recent global estimate of estuarine CH$_4$ emissions is the highest because it accounts for the direct emissions of CH$_4$ from sediment to atmosphere (when inter-tidal areas are exposed) (Borges and Abril, 2011). Yet, published estuarine CH$_4$ emissions are most probably

under-estimated because they do not account for CH$_4$ ebullition and gas flaring, although emissions to the atmosphere of CH$_4$ originating from gas-rich sediments in coastal environments have been shown to be intense (Borges et al., 2016; 2017). Reported CO$_2$ and CH$_4$ emissions from rivers are also highly uncertain and the reported values also span a considerable range. Global riverine CO$_2$ emission

estimates range between 0.1 PgC yr$^{-1}$ (Liu et al., 2010) and 1.8 PgC yr$^{-1}$ (Raymond et al., 2013), while riverine CH$_4$ emission estimates range between 2 TgCH$_4$ yr$^{-1}$ (Bastviken et al., 2011) and 27 TgCH$_4$ yr$^{-1}$ (Stanley et al., 2016). Both CO$_2$ and CH$_4$ riverine emissions mainly occur in tropical areas (Borges et al., 2015a,b).

     The first studies of CO$_2$ and CH$_4$ dynamics and emissions from estuaries were

carried out during the late 1990's in Europe (Frankignoulle et al., 1996; 1998; Middelburg et al., 2002) and the USA (Cai and Wang, 1998). Since then, CO$_2$ data coverage has tremendously increased with additional studies at sub-tropical and tropical latitudes (for example Sarma et al., 2012; Chen et al., 2012; Rao and Sarma, 2016) and in the large river-estuarine systems such as the Amazon (Lefèvre et al.,

2017), the Mississippi (Huang et al., 2015), the Changjiang (Yangtze) (Zhai et al., 2007; Zhang et al., 2008), the Pearl (Guo et al., 2009). The number of studies on CH$_4$ in estuarine and coastal environments has not increased in recent years as spectacularly as those concerning CO$_2$, attracting less research efforts because the marine source of CH$_4$ to the atmosphere (0.4-1.8 TgCH$_4$ yr$^{-1}$, Bates et al., 1996;

Rhee et al., 2009) is very modest compared to other natural and anthropogenic CH$_4$ emissions (Kirschke et al., 2013). Continental shelves and estuaries are more intense sources to the atmosphere of CH$_4$ than the open ocean, in particular shallow and permanently well-mixed coastal zones (Borges et al., 2016; 2017). Yet, numerous large river-estuarine systems remain totally uncharted with respect to CO$_2$

and CH$_4$ data, such as the Mekong although it is the World's 10[th] largest river in water discharge (470 km$^3$ yr$^{-1}$), 12[th] largest in length (4,800 km), and 21[st] largest in drainage area (795,000 km$^2$) (Li and Bush, 2015).

     As a contribution to the special issue in *Biogeosciences* on "Human impacts on carbon fluxes in Asian river systems", we report a data-set obtained in the three



branches (My Tho, Ham Luong, Co Chien) of the Mekong delta (Fig. 1) in December 2003, April 2004, and October 2004 of biogeochemical variables related to carbon cycling (pH, total alkalinity (TA), $O_2$, calculated partial pressure of $CO_2$ (p$CO_2$), dissolved $CH_4$ concentration, particulate (POC) and dissolved (DOC) organic carbon

concentration and stable isotopic (SI) composition, particulate nitrogen (PN), dissolved inorganic carbon (DIC) SI composition, total suspended matter (TSM)). The aim of the paper is to give a general description of carbon cycling in the Mekong delta estuarine system, that can be used as a reference state to evaluate future changes in response to changes in hydrology related the construction of planned

large dams (leading to water abstraction and sediment retention), eutrophication, shoreline erosion, and sea-level rise.

## 2.   Material and methods

### 2.1.   Description of the Mekong River and Delta

Himalayan rivers (Yangtze, Mekong, Salween, Ayeyarwady, Ganges, Brahmaputra, Ganges, Indus) are among the World's largest. The Mekong River is one of the longest rivers among the Himalayan watersheds, ranking it 12th longest

river in the World. It flows 4,800 km from the eastern part of the Tibetan Plateau through six different countries (China, Myanmar, Lao People's Democratic Republic (PDR), Thailand, Cambodia, Vietnam), into the South China Sea, draining an area of 795,000km². The basin is divided into the Upper Mekong (parts of China and Myanmar, surface of 195,000 km², first 2,000 km in length), and the Lower Mekong

(parts of Lao PDR, Thailand, Cambodia and Vietnam, surface of 600,000 km²). The Upper Mekong is mountainous (altitude 400-5,000 m) with no significant large tributaries and a low population density (<10 inhabitants km$^{-2}$). The Lower Mekong is lowland, drains very large tributary river systems, and is densely populated (80-460 inhabitants km$^{-2}$). Climate ranges from cold temperate in the Upper Mekong to

tropical monsoonal in the Lower Mekong. The annual flow of the Mekong River is ~470km³, ranking 10th among the World largest rivers (Dai and Trenberth, 2002). Water source is snowmelt in the Upper Mekong, and surface runoff in the Lower Mekong. Seasonal variations in freshwater flow are controlled by the East Asian monsoons, resulting in an annual unimodal flood pulse. About 75% of the annual flow




occurs in four months (July-October). The annual sediment load was ~130-160 million tons in the 1960's and 110 million tons in the 1990's (Milliman and Farnsworth, 2011). The solute annual transport is 123 million tons (Gaillardet et al., 1999). Exposed lithological strata are dominated by shales (43.2%), followed by carbonates (21.4%), shield rocks (18.2%), sands and sandstone (8.4%), basalts (5.8%) and acid volcanic rocks (2.9%) (Amiotte Suchet et al., 2003). The Mekong River basin is populated by 70 million people and this population is expected to increase to 100 million by 2050 (Varis et al., 2012). Recent and fast economic development has substantially increased the use of water resources (Piman et al., 2013), in particular for agriculture, energy (hydropower), and fishery (Västilä et al., 2010). Until recently, the Mekong River was considered one of the last unregulated great rivers with a flow regime close to its natural state (Adamson et al., 2009). Economic development in the region has led to the construction of several dams mainly for the production of hydropower, potentially affecting water and sediment flows (Fu et al., 2008; Wang et al., 2011; Lu et al., 2014; Piman et al., 2013; 2016). The construction of major infrastructures is planned on the transboundary Srepok, Sesan and Srekong Rivers, which contribute up to 20% of the total annual water flow of the Mekong (Piman et al., 2016).

The Mekong River delta covers an area of 50,000 km$^2$ and is the third largest tide-dominated delta in the World after the Amazon and Ganges-Brahmaputra deltas. It is tremendously important in the food supply and economic activity of Vietnam, as it sustains 90% of rice (>20 million tons annually) and 60% of seafood national production. The development of shrimp farming in the delta has led to the reduction of mangrove forests (de Graaf and Xuan, 1998; Nguyen et al., 2011) that nowadays only remain significantly in the Ca Mau Province. Shrimp farming started in the late 1970's, accelerated during the mid-1980's until present (de Graaf and Xuan, 1998; Tong et al. 2010). The delta is populated by more than 17 million people (>80% in rural areas), representing nearly a quarter of Vietnam's total population, with an annual population growth of more than 2%. The delta is a low-lying area with an average elevation of < 2 m above sea level, making it one of the most vulnerable deltas in the World to sea level rise (IPCC, 2014). The decrease in freshwater and sediment delivery combined to the rising sea-level and subsidence, as well as coastal (shoreline) erosion are potential threats for economic activities in the Mekong delta, for instance due to the impact of salinity intrusion on agriculture, compromising



economy and livelihood of local populations (Smajgl et al., 2015). Several studies predict that a large fraction (70-95%) of the sediment load could be trapped by hydropower reservoirs if all of the planned infrastructures are effectively build (Kummu et al., 2010; Kondolf et al., 2014). In addition, sediment river delivery could

also change in response to changes in climate (Västilä et al. 2010; Lauri et al., 2012; Darby et al., 2016). This would have important consequences on the sediment deposition in the delta that seems to have already shifted from a net depositional (accretion) regime into a net erosion regime (Anthony et al., 2015; Liu et al., 2017).

**2.2.   Sampling**

Sampling in the three branches of the Mekong delta (My Tho, Ham Luong, Co Chien, Fig. 1) was carried during three field campaigns (29/11/2003-05/12/2003; 02/04/2004-07/04/2004; 14/10/2004-19/10/2004) on the inspection boat of the Bến

Tre Fishery Department, in collaboration with the Research Institute for Aquaculture N°2 (Ho Chi Minh City). Sampling in the mangrove creeks of the Ca Mau province was carried during two field campaigns (10/04/2004-14/04/2004; 23/10/2004-25/10/2004) with a speed boat. The map of the sampling stations in the mangrove creeks of the Ca Mau province is given by Koné and Borges (2008) who reported

$pCO_2$, $\%O_2$ and TSM data.

pH, TA, $O_2$, TSM, POC and $\delta^{13}$C-POC, PN, $\delta^{13}$C-DIC were collected and analysed at all stations of all three field campaigns. Dissolved $CH_4$ concentration was collected during the two last field campaigns, DOC during the last field campaign, and dissolved silica (DSi) during the second field campaign.

**2.3.   Sample collection and analysis**

Salinity and water temperature were measured in-situ using a portable thermosalinometer (WTW Cond-340) with a precision of $\pm0.1$ and $\pm0.1°C$,

respectively. Subsurface waters (top 30 cm) were sampled with a 1.7 L Niskin bottle (General Oceanics) for the determination of pH and dissolved gases sampled with a silicone tube. Water for the determination of $O_2$ was sampled in a Winkler type borosilicate bottle and the oxygen saturation level ($\%O_2$) was measured immediately after collection with a polarographic electrode (WTW Oxi-340) calibrated on saturated

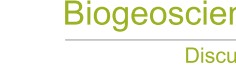


air, with an accuracy of ±0.1%. pH was also sampled in a Winkler type of bottle and measured immediately after collection with a combination electrode (Metrohm 6.0232.100) calibrated on the U.S. National Bureau of Standards scale as described by Frankignoulle and Borges (2001), with a precision and estimated accuracy of

respectively ±0.001 and ±0.005 pH units. Water for the determination of $CH_4$ was sampled in 50 ml borosilicate serum bottles poisoned with 100 µl of a saturated solution of $HgCl_2$ sealed with a butyl stopper and crimped with an aluminium cap. The $CH_4$ concentration was measured by the headspace technique (Weiss 1981) using a gas chromatograph (GC) with flame ionization detection (GC-FID, Hewlett Packard

HP 5890A), calibrated with certified $CH_4$:$N_2$ mixtures of 10 and 200 ppmv $CH_4$ (Air Liquide, France), with a precision of ±5%. Water for the analysis of $\delta^{13}$C-DIC was sampled in 12 mL Exetainer vials and poisoned with 20 µL of a saturated $HgCl_2$ solution. A He headspace was created, and ~300 µL of $H_3PO_4$ was added to convert all DIC species to $CO_2$, and after overnight equilibration, part of the headspace was

injected into the He stream of an Elemental Analyzer – Isotope Ratio Mass Spectrometer (EA-IRMS; ThermoFinnigan Flash1112 and ThermoFinnigan Delta+XL) for $\delta^{13}$C measurements, with a precision of better than ± 0.2 ‰.

Samples for TSM were filtered on pre-weighed and pre-combusted (5 h at 450°C) 47 mm Whatman GF/F filters, rinsed with mineral water to avoid salt contributions,

and subsequently dried. Samples for POC, PN, and $\delta^{13}$C$_{POC}$ were filtered on pre-combusted 25 mm Whatman GF/F filters and dried. These filters were later decarbonated with HCl fumes under partial vacuum for 4 h, re-dried and packed in Ag cups. POC and PN were determined on a ThermoFinnigan Flash EA1112 using acetanilide as a standard, and the resulting $CO_2$ was measured on a ThermoFinnigan

delta+XL interfaced via a ConfloIII to the EA. Reproducibility of $\delta^{13}$C$_{POC}$ measurements was better than ± 0.2 ‰. Samples for DOC and $\delta^{13}$C$_{DOC}$, TA, DSi, major cations ($Ca^{2+}$, $Mg^{2+}$, $Na^+$, $K^+$) were obtained by pre-filtering water on cellulose acetate filters for DSi, and pre-combusted Whatman GF/F filters for the other variables, followed by filtration on 0.2 µm cellulose acetate syringe filters (Sartorius).

DOC and $\delta^{13}$C$_{DOC}$ were stored in 40 ml borosilicate bottles and preserved by addition of 50 µL of $H_3PO_4$, DSi and major cations were stored in 20 ml high density polyethylene scintillation vials and preserved with $HNO_3$ (50 µl from DSi, 10 µl for major cations), and TA was stored un-poisoned in 100 ml polyethylene vials. DOC concentrations and $\delta^{13}$C signatures were measured with a modified Thermo



HiperTOC TOC-analyzer, interfaced with a Thermo delta +XL IRMS as described by Bouillon et al. (2006). DSi was measured with the colorimetric method of Koroleff (1983), with a precision of ±0.1 µmol $L^{-1}$. TA was measured on 50 ml samples by automated Gran titration with 0.1 M HCl as titrant, with a reproducibility of ±1 µmol

$kg^{-1}$. Samples for major cations were measured by inductively coupled plasma – atomic emission spectrometry (ICP-AES) and with a reproducibility better than ±3 %.

Measurements of TA and pH were used to compute $pCO_2$ and DIC using the carbonic acid thermodynamic dissociation constants of Cai and Wang (1998), with an estimated accuracy of ±5 % and ±5 µmol $kg^{-1}$, respectively (Frankignoulle and

Borges, 2001).

Air-water fluxes of $CO_2$ ($FCO_2$) and $CH_4$ ($FCH_4$) were calculated according to:

$$F = k.\Delta G \tag{1}$$

where $F$ is the flux of the gas, $\Delta G$ is air-water gradient of the gas and $k$ is the gas transfer velocity.

Values of $k$ were computed using wind speed field measurements with a handheld anemometer, and the parameterization as a function of wind speed given by Raymond and Cole (2001) (the "non-dome" parameterisation). The $k$ values in

estuarine environments are highly variable and parameterizations as a function of wind speed are site-specific due to variable contribution of fetch limitation and tidal currents (Borges et al., 2004). The parameterization of Raymond and Cole (2001) probably provides minimal $k$ values, so the $FCO_2$ and $FCH_4$ values given hereafter are considered to be conservative estimates. Atmospheric $pCO_2$ values were

retrieved from the National Oceanic and Atmospheric Administration Earth System Research Laboratory atmospheric measurement network data-base at station Guam (Mariana Islands, 13.386°N 144.656°E), located in the Pacific Ocean, approximately at the same latitude as Mekong delta. For $CH_4$, a constant atmospheric value of 1.8 ppm was used. The Henry constant of $CO_2$ and $CH_4$ was computed from salinity and

temperature according to Weiss (1974) and Yamamoto et al. (1976), respectively, and the Schmid number for $CO_2$ and $CH_4$ was computed from temperature according to Wanninkhof (1992). The air-water $CO_2$ and $CH_4$ values were area-averaged and scaled to the surface of the three estuarine branches using surface areas derived from satellite images with Google Earth.



### 2.4. Mixing models

We used a mixing model for TA, DIC and $O_2$ that assumes conservative mixing and
5    no gaseous exchange with the atmosphere for a solute (E), according to:

$$E_S = \left( \frac{E_M - E_F}{Sal_M - Sal_F} \right) Sal + E_f \tag{2}$$

where $E_S$ is the concentration of E at a given salinity (=Sal), $E_F$ is the concentration of
10    E at the freshwater end-member (with a salinity of $Sal_F$), $E_M$ is the concentration of E
at the marine end-member (with a salinity of $S_M$).
The conservative mixing of $\delta^{13}C_{DIC}$ was computed according to Mook and Tan
(1991):

$$\delta^{13}C \cdot DIC = \frac{Sal(DIC_F \delta^{13}C \cdot DIC_F - DIC_M \delta^{13}C \cdot DIC_M) + Sal_F DIC_M \delta^{13}C \cdot DIC_M - Sal_M DIC_F \delta^{13}C \cdot DIC_F}{Sal(DIC_F - DIC_M) + Sal_F DIC_M - Sal_M DIC_F} \tag{3}$$

where Sal is the salinity of the sample, $DIC_F$ and $\delta^{13}C_F \cdot DIC$ are, respectively, the DIC
concentration and stable isotope composition at the freshwater end-member, $DIC_M$,
and $\delta^{13C}C_M \cdot DIC$ are, respectively, the DIC concentration and stable isotope
composition at the marine end-member.

### 3. Results and discussion

### 3.1. Spatial and seasonal variations in the main branches of the Mekong delta (My Tho, Ham Luong, Co Chien)

The three sampling cruises covered three distinct phases of the hydrological cycle
(Fig. 2): low water (April 2004), close to high water (October 2004), and falling water
(December 2003). This strongly affected the salinity intrusion into the three inner
estuarine channels (My Tho, Ham Luong, Co Chien): in December 2003 and October
30    2004, freshwater was observed throughout the inner estuarine channels down to the
mouths, while in April 2004, the salinity intrusion occurred up to 60km upstream of
the estuarine mouths (Fig. 3). The $pCO_2$ values showed a general inverse pattern
compared to salinity and strongly decreased offshore from the mouth of the three
delta arms in December 2003 and October 2004, while the decreasing pattern of





$pCO_2$ occurs within the three estuarine channels in April 2004. In December 2003 and October 2004, the $pCO_2$ values upstream (freshwater) ranged between 1,895 and 2,664 ppm, well above atmospheric equilibrium (362 ppm), and above the range of values (703-1,597 ppm) reported by Alin et al. (2011) in the upstream reaches of the Mekong the river network during the high water period (September-October 2004-2005). This difference might be due to a stronger human influence on the densely populated Mekong delta, or to geomorphology (lowland rivers versus higher altitude rivers). The $pCO_2$ values from the extensive dataset in the Mekong River at Tan Chau (~100 km upstream of the delta) ranged between 390 and 4861 ppm and averaged 1325 ppm (Li et al., 2013), encompassing the $pCO_2$ values we obtained in the freshwater part of the delta. The high $pCO_2$ values in freshwaters in December 2003 and October 2004 corresponded to low $\%O_2$ values (69-84%) indicative of degradation of organic matter. In April 2004, the most upstream sampled stations of the delta (freshwater) were characterised by $pCO_2$ values (479-753 ppm) closer to atmospheric equilibrium and high $\%O_2$ values (98-106%) indicative of freshwater phytoplankton development during low water, probably related to an increase of water residence time related to low freshwater discharge (Reynolds and Descy, 1996), as also observed in other tropical rivers (for example Descy et al., 2017). Phytoplankton development during low water was also reported in the Upper Mekong River (confluence with the Tonle Sap River) by Ellis et al. (2012), based on elemental and lignin analyses. The impact of biological activity on $CO_2$ dynamics in the uppermost freshwater part of the estuary, was confirmed by $\delta^{13}C$-DIC values that were higher in April 2004 compared to December 2003 and October 2004. Indeed, $pCO_2$ was positively related to freshwater discharge, while $\%O_2$ and $\delta^{13}C$-DIC were negatively related to freshwater discharge (Fig. 4), as also shown in other tropical rivers such as the Oubangui (Bouillon et al., 2012; 2014). The dataset in the Mekong River at Tan Chau reported by Li et al. (2013), shows a similar seasonal pattern, with lower $pCO_2$ values during low water (March-May) and higher $pCO_2$ values during high water (October-December). In April 2004, there was a marked increase of $pCO_2$ from the most up-stream stations (salinity 0) to the stations located at 60 km from Vĩnh Long (corresponding roughly to a salinity of 2). This increase of $pCO_2$ was mirrored by a general decrease of $\%O_2$, suggesting enhanced organic matter degradation in the oligohaline estuarine region, typical of estuarine environments (for example, Morris et al., 1978; Bianchi, 2006). In parallel, there was a general increase




of DSi from salinity 0 to 2 suggesting that part of the enhanced organic matter degradation in the upper estuary in April 2004 was fuelled by the decay of freshwater diatoms due to haline (osmotic) stress (for example, Muylaert and Sabbe, 1999; Ragueneau et al., 2002), as also observed in other tropical estuaries such as the

5     Tana and the Kidogoweni (Bouillon et al., 2007a,b). In December 2003 and April 2004, a general gradual increase of $pCO_2$ was also observed along the estuarine channels towards the mouth, although the $\%O_2$ decrease was less marked than in April 2004. The TA values at zero salinity ranged from ~960 to ~980 µmol kg$^{-1}$ in October 2004 and December 2003, respectively, lower than in April 2004 (~1,400

10    µmol kg$^{-1}$). These values are higher than the $HCO_3^-$ concentration of 949 µmol kg$^{-1}$ reported by Meybeck and Carbonnel (1975) at Phnom Penh from January 1961 to 1962. The data of Meybeck and Carbonnel (1975) were obtained about 230 km upstream of our sampling sites in the Mekong delta, so the difference could be due to the general downstream increase in dissolved ions typically observed in rivers (for

example, Whitton 1975), but we cannot exclude methodological differences, or long-term changes. Li and Bush (2015) did not identify clear long-term trends in $HCO_3^-$ at two stations in the Lower Mekong river from 1960 to 2011. Our TA values converge with the median (1082 µmol kg$^{-1}$) of a large data-set during 1972-1996 period from 42 stations in the lower Mekong delta compiled by the Mekong River Commission and

reported by Li et al. (2014). The seasonal variations of TA follow those of freshwater discharge (Fig. 4), as also shown in other major rivers such as the Mississippi (Cai et al., 2008) and the Oubangui (Bouillon et al., 2012; 2014). TA in freshwater was correlated to $Ca^{2+}$ with a slope of 2.0 (Fig. S1) consistent with the weathering of calcite ($CaCO_3$, $HCO_3^-$:$Ca^{2+}$ = 2:1) and to $Mg^{2+}$ with a slope of 2.2 consistent with the

weathering of dolomite (($Ca,Mg)CO_3$, $HCO_3^-$:($Ca^{2+}$,$Mg^{2+}$) = 2:1). Such stoichiometric ratios between $HCO_3^-$ and $Ca^{2+}$ and $Mg^{2+}$ might also result from weathering of silicate rocks such as anorthite (Ca-plagioclase feldspar, $CaAl_2Si_2O_8$, $HCO_3^-$:$Ca^{2+}$ = 2:1), chlorite ($Mg_5Al_2Si_3O_{10}$, $HCO_3^-$:$Mg^{2+}$ = 2:1) or olivine ($MgSiO_4$, $HCO_3^-$:$Mg^{2+}$ = 2:1). However, Li et al. (2014) have shown based on an extensive water chemistry dataset

that carbonate rock weathering largely dominates silicate weathering in the Lower Mekong River, and this seems to be also the case in the Upper Mekong River (Manaka et al., 2015). TA in freshwater was also correlated to $Na^+$ but with a slope of 0.5, lower than expected from the weathering of Albite ($NaAlSi_3O_8$; $HCO_3^-$:$Na^+$ = 1:1), and to $K^+$ but with a slope of 14, higher than expected from the weathering of



microcline (K-Feldspar, $KAlSi_3O_8$, $HCO_3^-$:$K^+$ = 1:1). Weathering of calcite alone would not account for all of the TA, but this would be the case for a mixture of weathering of calcite and dolomite (Fig. S2), also in agreement with the analysis of Li et al. (2014).

As a function of salinity, $pCO_2$ and $\%O_2$ showed in the three delta channels,
regular decreasing and increasing patterns, respectively (Fig. 5). The lowest off-shore $pCO_2$ value was observed in October 2004 (314 ppm at 27.0 salinity), lower than in December 2003 (509 ppm at 17.9 salinity) and April 2004 (423 ppm at 31.9 salinity). TA showed a linear evolution against salinity, indicative of near conservative mixing behaviour. This was consistent with a near conservative mixing behaviour of
major cations ($Ca^{2+}$, $Mg^{2+}$, $K^+$, $Na^+$) (Fig. S3). DIC generally followed the seasonal and spatial patterns of those of TA. $\delta^{13}C$-DIC showed a typical increasing pattern with salinity (Mook and Tan 1991; Bouillon et al., 2012), resulting from the mixing of freshwater with more negative $\delta^{13}C$ signatures (-14 to -8 ‰) and marine water with a $\delta^{13}C$ signature close to 0‰. The $^{13}C$-depleted signature in freshwater DIC results
mainly from the degradation of organic matter, which contributes $CO_2$ with a signature close to that of the source organic carbon which in the Mekong delta for POC ranged between -28 and -26 ‰, and from the weathering of carbonate and silicate minerals (which are typically driven by reaction with $CO_2$ derived from organic matter). $CH_4$ showed very strong seasonal variations in freshwaters of the Mekong
delta with values < 20 nmol $L^{-1}$ in April 2004 and values ranging between 25 and 220 nmol $L^{-1}$ in October 2004. The seasonal $CH_4$ variation could result from the flooding of riparian and floodplain areas and/or $CH_4$ inputs from surface run-off during the rainy season and high water period leading to high $CH_4$ values in October 2004. The downstream decrease of $CH_4$ in the estuarine salinity mixing zone is typical (Borges
and Abril 2012; Upstill-Goddard and Barnes, 2016), resulting from $CH_4$ riverine loss in the estuary due to emission to the atmosphere, microbial $CH_4$ oxidation and mixing with marine waters that have $CH_4$ concentrations close to atmospheric equilibrium (Rhee et al., 2009).

TSM values showed marked spatial gradients in October 2004 with high values
up to 447 mg $L^{-1}$ in freshwaters and very low values (2 mg $L^{-1}$) in marine waters. In April 2004 and December 2003, TSM values in freshwaters were lower and the spatial variations along the salinity gradient were less obvious. POC concentration ranged between 0.2 and 4.0 mg $L^{-1}$, and the seasonal and spatial variations of POC were very similar to those of TSM. %POC values ranged between 2 and 4% typical



for the corresponding range of TSM values in World rivers (Meybeck, 1982; Ludwig et al., 1996) and in estuaries (Abril et al., 2002), and within the range measured in the lower Mekong just upstream of the confluence with the Tonle Sap river during an annual cycle by Ellis et al. (2012). However, %POC values were distinctly higher (up to ~13%) in marine waters in October 2004 probably resulting from a phytoplankton bloom, as also testified by low POC:PN ratios (as low as 4.9), high %$O_2$ (up to 114%) and $\delta^{13}$C-DIC (up to 0‰) values, and low $pCO_2$ (as low as 232 ppm) values. The phytoplankton probably resulted from higher light availability in marine waters owing to lower TSM values (down to 2 mg $L^{-1}$). $\delta^{13}$C values of POC in the freshwater part of the delta (salinity <1) from the 3 sampling campaigns averaged -26.7 ± 0.7 ‰ (n=34), distinctly higher than the data from Ellis et al. (2012) which averaged -29.8 ± 0.9 ‰, but similar to data collected by Martin et al. (2013; average -26.4 ‰) at the same site as the Ellis et al. (2012) study. These $\delta^{13}C_{POC}$ values are consistent with the expected dominance of terrestrial C3 vegetation in the riverine organic carbon load.

In October 2004, DOC showed a decreasing pattern while $\delta^{13}$C-DOC values decreased, as typically observed in estuaries (Bouillon et al., 2012). Within the freshwater zone (salinity <1), $\delta^{13}C_{DOC}$ values (-27.8 ± 0.6 ‰, n=19) are again consistent with a dominance of terrestrial C3 vegetation inputs, and close to values reported by Martin et al. (2013) slightly upstream in the lower Mekong. $\delta^{13}$C values were lower in DOC than POC for the same samples in October 2004 (Fig. 6), probably reflecting the more refractory nature of riverine DOC compared to POC, the latter being removed faster during estuarine mixing, being gradually replaced by POC of phytoplankton origin with a higher $\delta^{13}$C value.

## 3.2. Distinct patterns in side channels compared to the main branches of the Mekong delta

The sampled biogeochemical variables showed distinct patterns in the side channels of the Mekong delta compared to the main channels (My Tho, Ham Luong and Co Chien), irrespective of the sampling period. The observed patterns are consistent with the influence from the very extensive ponds devoted to shrimp farming that border the side channels of the Mekong delta (Tong et al. 2010). TSM, POC and %POC values were higher in the side channels than in the three main estuarine channels. The DOC concentrations from the October 2004 cruise were also



higher in the side channels. Higher %POC values could indicate a higher contribution of phytoplankton biomass to TSM, and this is consistent the $\delta^{13}$C-POC values that are about 5-6 ‰ lower than the values in the three main estuarine channels at the same salinity values. There is an isotopic fractionation by phytoplanktonic primary
production of about 20‰ during DIC uptake (Hellings et al., 1999), corresponding roughly to the difference in $\delta^{13}$C values between POC (overall average: -27.4±1.8‰) and DIC (overall average: -8.2±2.4‰) in the side channels. The phytoplankton primary production was probably sustained by high inorganic nutrients inputs from shrimp farming pods typically observed in adjacent channels (for example Cardozo
and Odebrecht, 2014) or within the ponds themselves (Alongi et al., 1999a). Yet, the more negative $\delta^{13}$C-DIC values in the side channels indicate sustained $CO_2$ production from organic matter degradation related to the shrimp ponds (Alongi et al., 2000), consistent with higher $pCO_2$ values and lower $%O_2$ in the side channels compared to the adjacent estuarine channels. The distinctly higher $CH_4$ values would
indicate that part of the organic matter degradation in the side channels occurs in sediments. Alongi et al. (1999b) showed that methanogenesis in the sediments of shrimp farming ponds themselves is low in the Ca Mau Province. This allows to suggest that the high $CH_4$ in the side channels were presumably coming from the side channels sediments and not from the shrimp farming ponds. The higher TA
values in the side channels than in estuarine channels could also indicate the effect of diagenetic anaerobic processes (for example, Borges et al., 2003). The concentrations of $Ca^{2+}$ and $Mg^{2+}$ did not show marked deviations as a function of salinity in the side channels compared to estuarine channels (Fig. S3), indicating TA generation was unrelated to dissolution of $CaCO_3$ or $CaMg(CO_3)_2$.
We further explored data using the difference (or anomaly) between observed data and data predicted from conservative mixing models, noted Δ (Fig. 7). Negative $\Delta\delta^{13}$C-DIC values were correlated to those of $\Delta O_2$ and ΔDIC, in particular in the side channels, as expected from production of $CO_2$ and consumption of $O_2$ due to degradation of organic matter. In October 2004, distinct positive $\Delta\delta^{13}$C-DIC were
associated to positive $\Delta O_2$ and negative ΔDIC in the Ham Luong and Co Chien resulting from high phytoplankton production in the most off-shore waters, as mentioned in the previous section. The relation between positive ΔDIC and negative $\Delta O_2$ in the side channels also indicates that degradation organic matter, while negative ΔDIC and positive $\Delta O_2$ indicate in October 2004 in the Ham Luong and Co

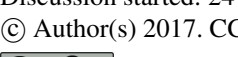


Chien confirm the occurrence of high phytoplankton production in the most off-shore waters. The slope of the linear regression of $\Delta DIC$ as a function of $\Delta O_2$ in the side channels ranged from 3.4 to 4.4. These values are distinctly higher than those expected from the degradation of organic matter following the Redfield stoichiometry

($\Delta DIC:\Delta O_2$ = 106:138 = 0.8). The slope of the relation between $\Delta DIC$ and $\Delta O_2$ in October 2004 in the Ham Luong and Co Chien (1.4) was lower than in the side channels but still higher than the one predicted from Redfield stoichiometry. One possible explanation is that the change of concentration due to the exchange of gases with atmosphere (equilibration) is faster for $O_2$ than $CO_2$ due to the effect on

the latter of buffer capacity of seawater. Another explanation that could explain the distinctly higher $\Delta DIC:\Delta O_2$ ratio in the side channels relates to anaerobic organic matter degradation in sediments that seems higher compared to estuarine channels as also suggested by higher $CH_4$ concentrations. The relative change of TA and DIC can be used to identify the processes involved in the generation of these quantities

(for example Borges et al., 2003). The slope of the linear regression of $\Delta TA$ versus $\Delta DIC$ ranged between 0.55 and 0.87, intermediary between the theoretical slopes for aerobic organic matter degradation (0.2) and sulfate-reduction (0.9), suggesting that TA and DIC were produced from the combination of these two processes. Such scenario is very likely, sulfate-reduction dominating in the sediments, and aerobic

respiration dominating in the water column. Our data does not allow to determine, whether these processes mainly occurred in the side channels or in the shrimp farming ponds themselves, although Alongi et al. (1999b) showed a strong dominance of aerobic respiration over other diagenetic degradation processes in sediments of shrimp ponds in Cau Mau Province. This would then allow to suggest

that sulfate-reduction was mostly occurring within the side channels. The $\Delta TA:\Delta DIC$ slope from the side channels correlated negatively to average salinity (Fig. 8) which is counter-intuitive since a higher contribution of sulfate-reduction ($\Delta TA:\Delta DIC$ ratio closer to 0.9) would have been expected at higher salinity (e.g. Borges and Abril 2011). This pattern might result from a higher aerobic respiration in the water column

of the side channels during the periods of low water (higher salinity), and/or a lower signal from sulfate-reduction occurring within the shrimp farming ponds. This scenario is consistent with the negative correlation between $\Delta O_2$ and salinity (Fig. 8). This could be due to higher exchange of water between the shrimp ponds and the adjacent channels during the low water (dry season) period.



### 3.3. Comparison with the Ca Mau mangrove creeks

The Ca Mau peninsula accounts for the largest proportion of mangrove forests in the Mekong Delta system. Data were gathered in two mangrove creek networks (Tam Giang and Kiên Vàng) allowing the comparison with data in the three estuarine channels of the Mekong delta (My Tho, Ham Luong and Co Chien) and associated side channels (hereafter referred to as Bến Tre Mekong delta, based on the name of the Province), where the bordering mangroves forests have been cleared for the implementation of shrimp farming ponds. Data comparison is limited to the April and October 2004 cruises (Fig. 9). $pCO_2$ was negatively related to $\%O_2$ in Ca Mau creeks and the Bến Tre Mekong delta owing to organic matter degradation as confirmed by the positive relation between $\Delta\delta^{13}C\text{-DIC}$ and $\%O_2$. However, data in the Ca Mau mangrove creeks allowed to expand the range of variations of $pCO_2$, $\%O_2$ and $\delta^{13}C\text{-DIC}$; the maximum $pCO_2$ value in the Ca Mau mangrove creeks was 6,912 ppm compared to 2,926 ppm in the Bến Tre Mekong delta; the minimum $\%O_2$ and $\delta^{13}C\text{-DIC}$ were, respectively, 37% and -14.6‰ in the Ca Mau mangrove creeks compared to 66% and -11.4‰ Bến Tre Mekong delta. As previously noted by Borges and Abril (2011), the variations of $pCO_2$ and $\%O_2$ in the Ca Mau mangrove creeks were related to the size of the creeks, the narrower and presumably shallower creeks being characterized by higher $pCO_2$ and lower $\%O_2$ values. As previous noted by Koné and Borges (2008), there were no significant seasonal variations of $\%O_2$ and $pCO_2$ in the Ca Mau mangroves creeks, despite the fact that salinity was highly variable among the two sampling cruises, on average 33.2 in April 2004 and 14.1 in October 2004, following the hydrological cycle (Fig. 2). The seasonal variations of $CH_4$ on the other hand were very marked, with much lower values in April 2004 (range 4-46 nmol $L^{-1}$, average 19 nmol $L^{-1}$) than in October 2004 (range 19-686 nmol $L^{-1}$, average 210 nmol $L^{-1}$). This is probably related to the salinity seasonal changes, the lowest $CH_4$ values corresponding to the highest salinities. We hypothesize that the increase of salinity leads to an increase of benthic sulfate-reduction due to the increase of $SO_4^{2-}$ availability, and a decrease of the transfer of $CH_4$ from sediments to the water column due to a partial inhibition of methanogenesis and/or an enhancement of anaerobic $CH_4$ oxidation. Such a hypothesis is consistent with the negative relationship in mangroves between sediment-air $CH_4$ fluxes and salinity (Borges and



Abril 2011). In October 2004, the $CH_4$ concentrations in the Ca Mau mangroves were general higher than in the Bến Tre Mekong delta three main channels, yet, the highest $CH_4$ concentrations were recorded in the side channels of the Bến Tre Mekong delta, most probably resulting from intense methanogenesis fuelled by high
organic matter loads from the shrimp farming ponds.

### 3.4.   $CO_2$ and $CH_4$ emissions to the atmosphere

As expected from the distribution of $pCO_2$, the $FCO_2$ values were higher in the
inner estuarine branches (My Tho, Ham Luong, Co Chien) than in the outer estuary (river plume) and the side channels. Although the $pCO_2$ in the side channels was higher than in the adjacent inner estuarine branches at similar salinities (Fig. 5), the overall $pCO_2$ within the inner estuarine branches was higher, owing to high values in the upper estuary. The seasonal variations of $FCO_2$ in the inner estuarine branches
followed the hydrological seasonal cycle, with the highest $FCO_2$ values in October 2004 during high water and the lowest $FCO_2$ values in April 2004 during low water (Table 1). The $FCO_2$ in the inner estuarine branches were well correlated to freshwater discharge (Fig. 10). This indicates that the $FCO_2$ seasonal variations are related to the riverine inputs either directly as $CO_2$ or as organic matter that can be
degraded within the estuary. During our cruises seasonal variations in water temperature were weak (range 26.7-31.5°C, on average 29.2°C), owing to the sub-tropical climate, consequently marked seasonality of $pCO_2$ and $FCO_2$ due to modulation of biological activity by water temperature does not occur unlike in temperate estuaries (for example Frankignoulle et al., 1998). The potential
contribution of riverine organic carbon and $CO_2$ inputs in sustaining estuarine $FCO_2$ was computed from freshwater discharge multiplied by POC and excess DIC (EDIC), respectively (EDIC is computed as the difference between observed DIC and DIC computed from TA and the atmospheric $pCO_2$ value, Abril et al., 2000). The average for the three cruises of riverine input of POC ($60 \times 10^6$ mol d$^{-1}$) and EDIC ($53 \times 10^6$ mol
d$^{-1}$) exceeded $FCO_2$ in the three estuarine branches ($53 \times 10^6$ mol d$^{-1}$), showing that these inputs were sufficient to sustain the $CO_2$ emissions from the estuary, and that part of the riverine POC and EDIC is transported to the outer estuary (river plume). $FCO_2$ in the side channels and outer estuary (or river plume) showed a less significant correlation with water discharge (Fig. 10), because other processes than



riverine inputs control $CO_2$ dynamics in these systems such as the inputs of carbon from the shrimp farming ponds for side channels, and primary production for the outer estuary. A phytoplankton bloom in the river plume in October 2004 explains why $F$CO$_2$ values were equivalent to those in December 2003, although freshwater
5    discharge was about two times lower.

The average $F$CO$_2$ in the inner estuarine branches of the Mekong delta (118
In April 2004, the $F$CH$_4$ values in the side channels of the Bến Tre Mekong delta were equivalent to those in the Ca Mau mangrove creeks, but were more than two
10    times higher in October 2004.

Differences of $F$CH$_4$ among the two 2004 cruises were very marked, with values in inner estuarine branches more than four times higher in October than April 2004. In April 2004, the $F$CH$_4$ values in the side channels of the Bến Tre Mekong delta were equivalent to those in the Ca Mau mangrove creeks, but were more than two
10    times higher in October 2004.

The average $F$CO$_2$ in the inner estuarine branches of the Mekong delta (118 mmol m$^{-2}$ d$^{-1}$) is higher than in the Pearl River inner estuary (46 mmol m$^{-2}$ d$^{-1}$, Guo et al., 2009) and the Yangtze River inner estuary (41 mmol m$^{-2}$ d$^{-1}$, Zhai et al., 2007), the two other major river systems bordering the East China Sea that have been
documented for $CO_2$ dynamics. The higher value in the Mekong is probably related to the dominance of freshwater in the inner estuary and low salinity intrusion within the estuary, related to the geomorphology (relatively narrow and linear estuarine channels, compared to the typical "funnel" shape estuarine morphology in the Yangtze and Pearl River estuaries). Indeed, the average salinity in the Pearl River
inner estuary was 17 (Guo et al., 2009), higher than the average salinity of 4 in the Mekong inner estuarine branches during our cruises. The average $F$CO$_2$ in the Ca Mau mangrove creeks (89 µmol m$^{-2}$ d$^{-1}$) was well within the range (-8-862 µmol m$^{-2}$ d$^{-1}$) and close to the average (63 µmol m$^{-2}$ d$^{-1}$) of $CO_2$ fluxes in mangrove estuarine creeks compiled globally by Call et al. (2015).

The $F$CH$_4$ seasonal variations within a given estuary and the $F$CH$_4$ variations from one estuary to another are notoriously large, so comparison of the $F$CH$_4$ in the Mekong delta with previously published studies is uneasy. The average $F$CH$_4$ value in the inner estuarine branches of the Mekong delta (118 µmol m$^{-2}$ d$^{-1}$) is within the range of values in European estuaries (17-1,352 µmol m$^{-2}$ d$^{-1}$) compiled by Upstill-
Goddard and Barnes (2016), but distinctly higher than the range of values for Indian estuaries (7-15 µmol m$^{-2}$ d$^{-1}$) reported by Rao and Sarma (2016). The $F$CH$_4$ in the Yangtze River estuary reported by Zhang et al. (2008) of 61 µmol m$^{-2}$ d$^{-1}$ is also higher than the range of $F$CH$_4$ in Indian estuaries. The $F$CH$_4$ in the Mekong delta inner estuarine branches was higher than the value in the Yangtze River estuary





probably because of the lower salinity intrusion into the Mekong delta (see above). The average $F$CH$_4$ in the Ca Mau mangrove creeks (160 µmol m$^{-2}$ d$^{-1}$) was well within the range (9-409 µmol m$^{-2}$ d$^{-1}$) and close to the average (283 µmol m$^{-2}$ d$^{-1}$) of CH$_4$ fluxes in mangrove estuarine creeks compiled globally by Call et al. (2015).

**Acknowledgements.** We are extremely grateful to the Research Institute for Aquaculture N°2 (Ho Chi Minh City) and the Bến Tre Fishery Department for logistical support during the collection of samples. Freshwater discharge data were kindly provided by Nguyen Hong Quang from the Vietnam National Satellite Center. This work was funded by the Fonds National de la Recherche Scientifique (FNRS) (1.5.066.03). AVB is a senior research associate at the FNRS.

**Author contribution.** AVB designed the experiment and carried out sample collection in the field. AVB, SB and GA analysed the samples, interpreted the data, and drafted the manuscript.

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



Table 1: Air-water fluxes of $CO_2$ ($FCO_2$ in mmol $m^{-2}$ $d^{-1}$) and $CH_4$ ($FCH_4$ in µmol $m^{-2}$ $d^{-1}$) in December 2003, April 2004 and October 2004, in the three inner estuarine branches of the Mekong delta (My Tho, Ham Luong and Co Chien), respective river plume and side channels, and in Cau Mau province mangrove creeks.

| | $FCO_2$ (mmol $m^{-2}$ $d^{-1}$) | $FCH_4$ (µmol $m^{-2}$ $d^{-1}$) |
|---|---|---|
| December 2003 | | |
|     Inner estuarine branches (IEB) | 122 | |
|     River plume (RB) | 56 | |
|     IEB+RB | 90 | |
|     Side channels | 85 | |
| April 2004 | | |
|     Inner estuarine branches | 105 | 43 |
|     River plume | 18 | 7 |
|     IEB+RB | 69 | 29 |
|     Side channels | 37 | 19 |
|     Ca Mau mangrove creeks | 61 | 22 |
| October 2004 | | |
|     Inner estuarine branches | 135 | 193 |
|     River plume | 44 | 46 |
|     IEB+RB | 70 | 87 |
|     Side channels | 88 | 701 |
|     Ca Mau mangrove creeks | 116 | 298 |
| Average of cruises | | |
|     Inner estuarine branches | 121 | 118 |
|     River plume | 39 | 26 |
|     IEB+RB | 76 | 58 |
|     Side channels | 70 | 360 |
|     Ca Mau mangrove creeks | 89 | 160 |



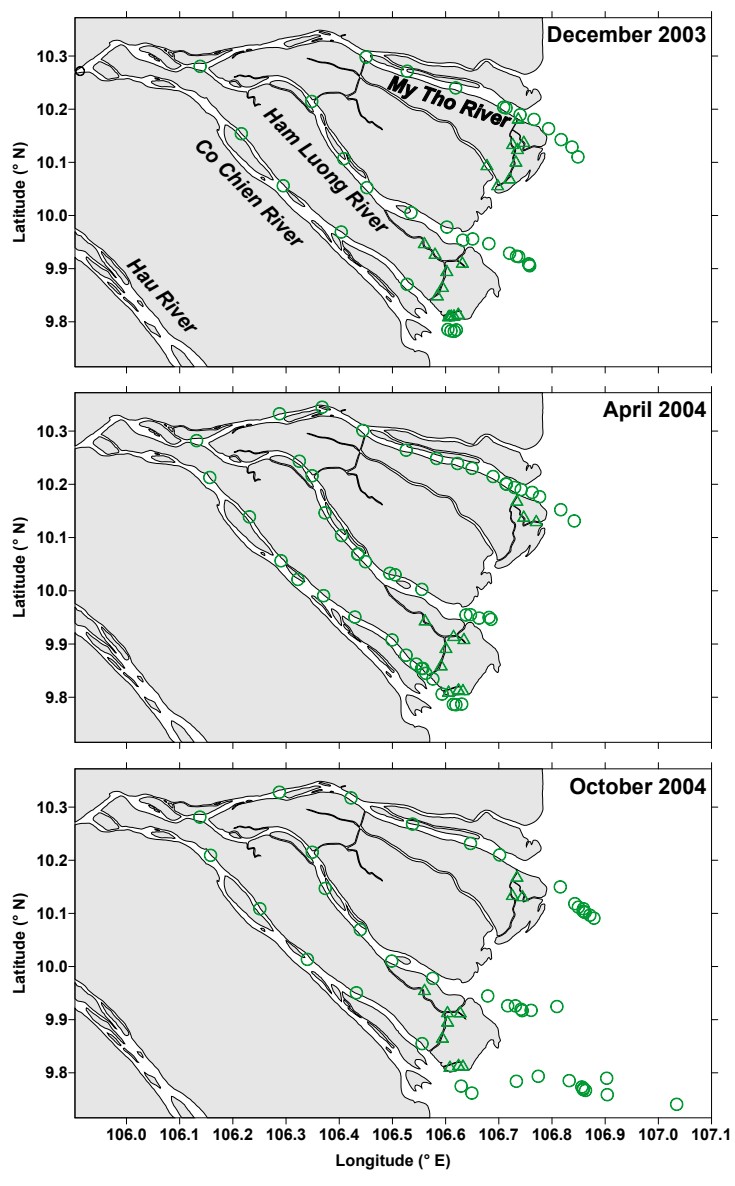

Fig.1 – Map of sampling stations in December 2003, April 2004 and October 2004, in the three inner estuarine branches of the Mekong delta (circles) (My Tho, Ham Luong and Co Chien) and side channels (triangles). Small black dot indicates the location of the bridge across the river at the city of Vĩnh Long from which the distance downstream is calculated in Figure 3.



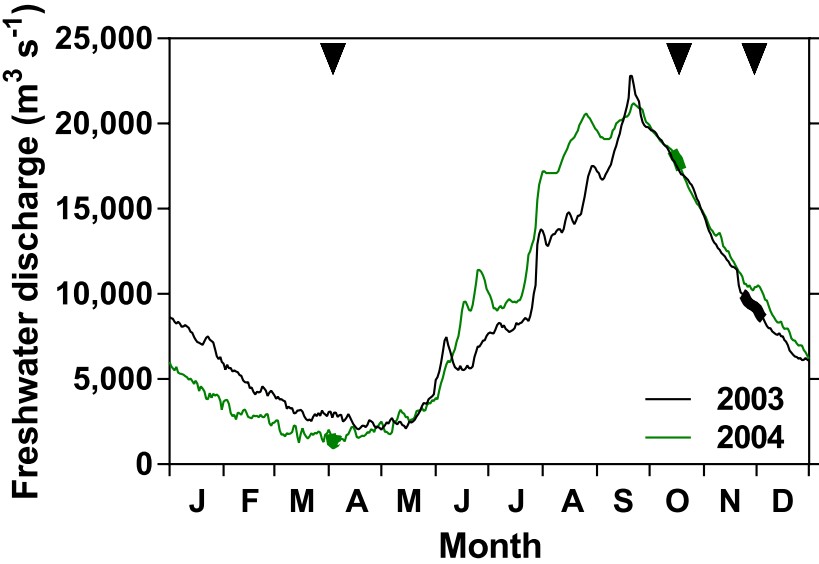

Fig. 2 – Seasonal evolution of daily freshwater discharge in the Mekong River at Tan Chau in 2003 and 2004. Thick lines and black triangles indicate the three sampling periods





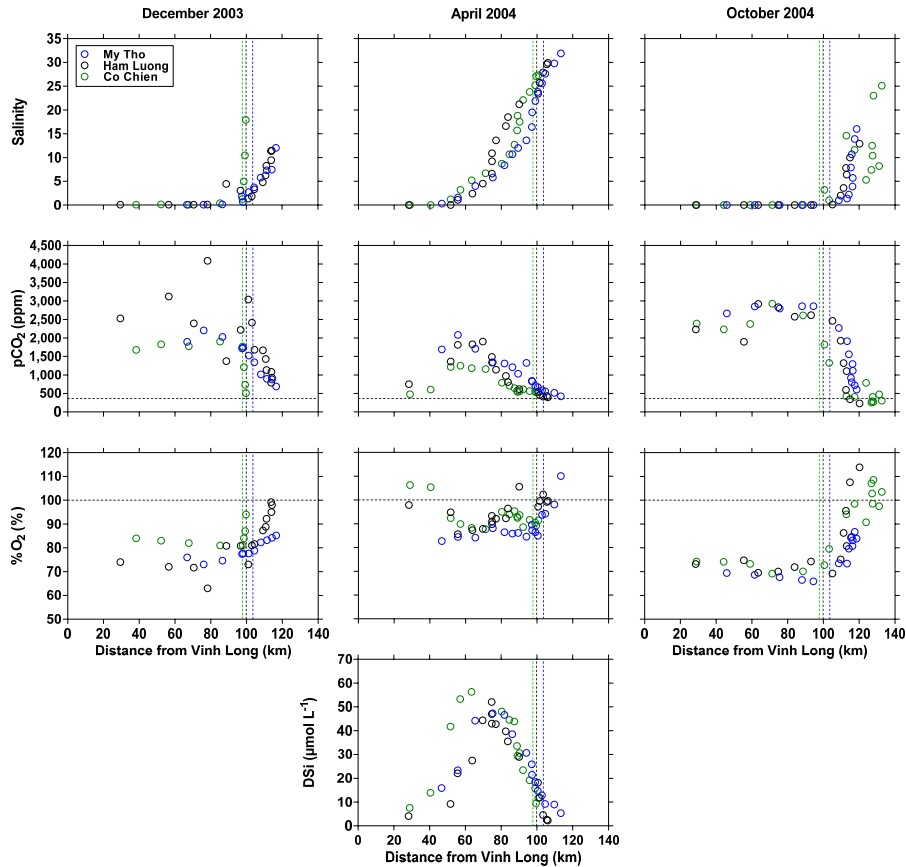

Fig. 3 – Distribution as a function of distance downstream of the city of Vĩnh Long of salinity, partial pressure of $CO_2$ (pCO$_2$ in ppm), oxygen saturation level (%O$_2$ in %), and dissolved silica (DSi in µmol L$^{-1}$) in the three branches of the Mekong delta (My Tho, Ham Luong and Co Chien), in December 2003, April 2004 and October 2004. The vertical dotted lines indicate the location of the river mouths.





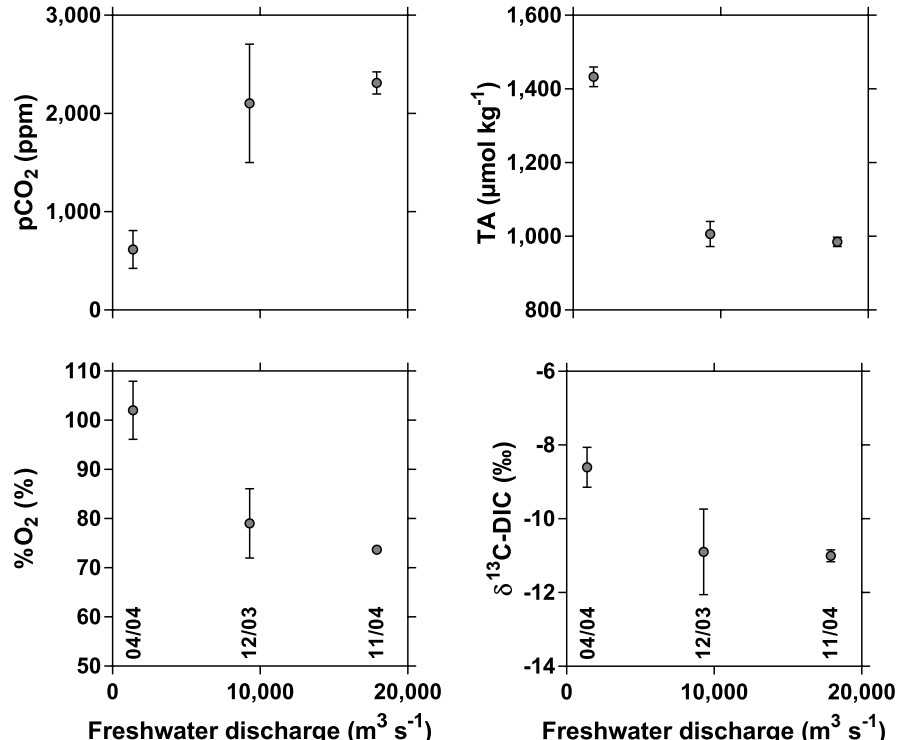

Fig. 4 – Variation as a function of freshwater discharge ($m^3\ s^{-1}$) of the partial pressure of $CO_2$ ($pCO_2$ in ppm), oxygen saturation level ($\%O_2$ in %), total alkalinity (TA in µmol $kg^{-1}$) and stable isotope composition of dissolved inorganic carbon ($\delta^{13}$C-DIC in ‰) in the freshwaters (salinity 0) of the three branches of the Mekong delta (My Tho, Ham Luong and Co Chien), in December 2003, April 2004 and October 2004. Sampling dates (MM/YY) are indicated in the bottom panels.



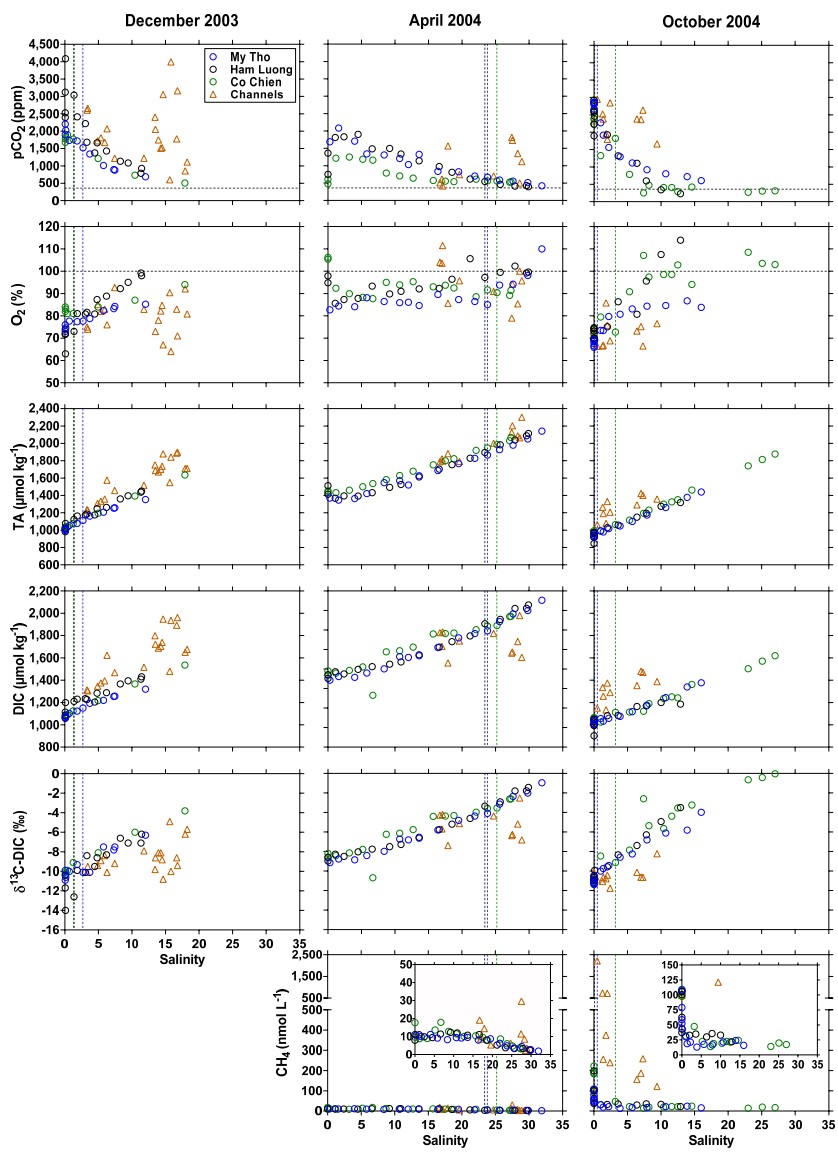

Fig. 5 - Distribution as a function of salinity of the partial pressure of $CO_2$ (pCO$_2$ in ppm), oxygen saturation level (%$O_2$ in %), total alkalinity (TA in µmol kg$^{-1}$), dissolved inorganic carbon (DIC in µmol kg$^{-1}$), stable isotope composition of DIC ($\delta^{13}$C-DIC in ‰), dissolved $CH_4$ concentration (nmol L$^{-1}$), total suspended matter (TSM in mg L$^{-1}$), particulate organic carbon (POC in mg L$^{-1}$), percent of POC in TSM (%POC in %), POC to particulate nitrogen ratio (POC:PN in mg:mg), stable isotope composition of POC ($\delta^{13}$C-POC in ‰), dissolved organic carbon (DOC in mg L$^{-1}$), and stable isotope composition of DOC ($\delta^{13}$C-DOC in ‰) in the three branches of the Mekong delta (My Tho, Ham Luong and Co Chien) and side channels, in December 2003, April 2004 and October 2004. The vertical dotted lines indicate the location of the river mouths.



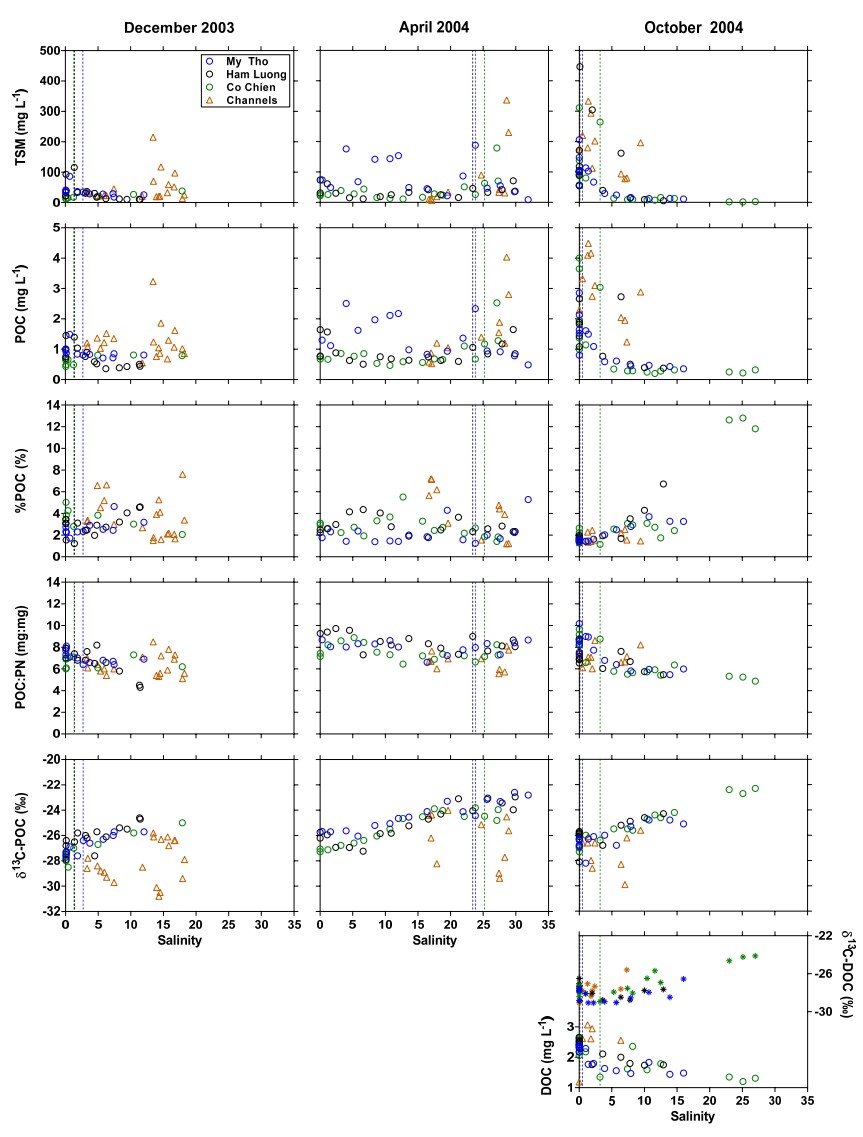

Fig. 5 (continued)





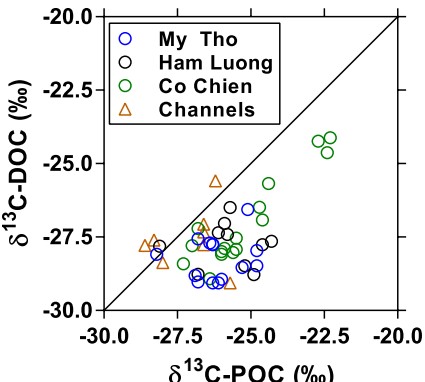

Fig. 6 : Stable isotope composition of dissolved organic carbon ($\delta^{13}$C-DOC in ‰) as a function of the stable isotope composition of particulate organic carbon ($\delta^{13}$C-POC in ‰) in the three branches of the Mekong delta (My Tho, Ham Luong and Co Chien) and side channels, in October 2004. The solid line indicates the 1:1 line.





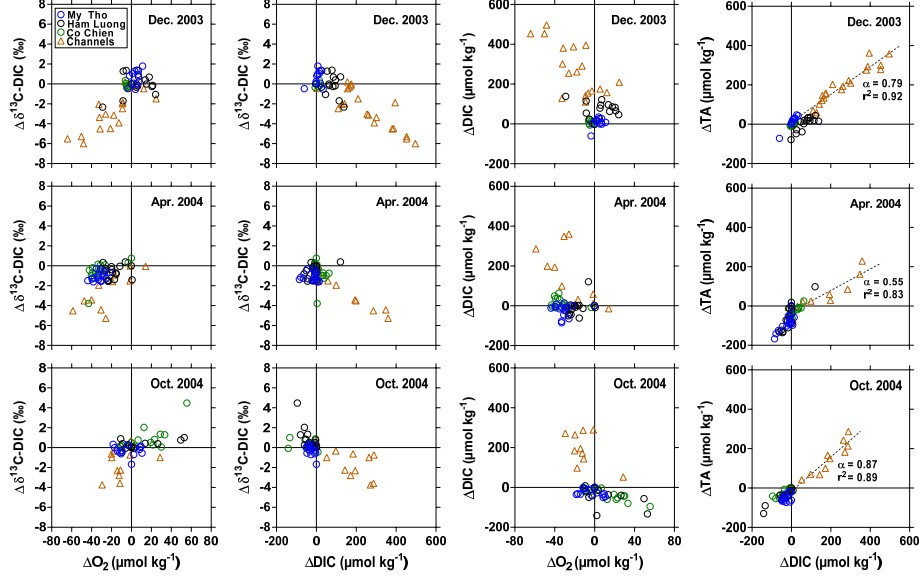

Fig. 7 – Deviations from conservative mixing lines of stable isotope composition of dissolved inorganic carbon (DIC) ($\Delta\delta^{13}$C-DIC in ‰) as a function of $O_2$ ($\Delta O_2$ in μmol kg$^{-1}$) and of DIC ($\Delta$DIC in μmol kg$^{-1}$), of $\Delta$DIC as a function of $\Delta O_2$, and of total alkalinity ($\Delta$TA in μmol kg$^{-1}$) as function of $\Delta$DIC, in the three branches of the Mekong delta (My Tho, Ham Luong and Co Chien) and side channels, in December 2003, April 2004 and October 2004. α indicates the slope of the linear regression line (dotted line).



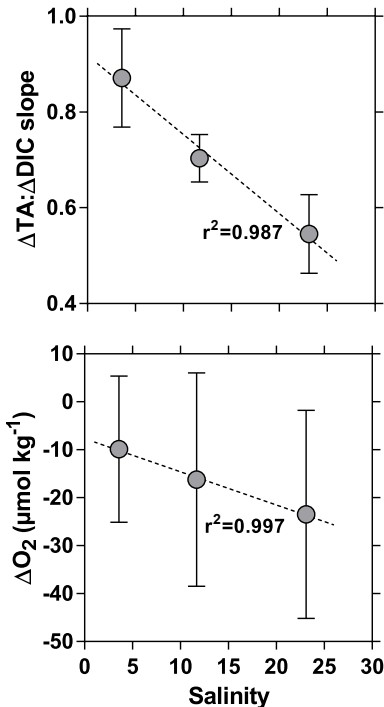

Fig. 8 – Variation as a function of salinity of the slope of regression line of the deviation from conservative mixing lines of total alkalinity (ΔTA in µmol kg$^{-1}$) and of dissolved inorganic carbon (ΔDIC in µmol kg$^{-1}$), of O$_2$ (ΔO$_2$ in µmol kg$^{-1}$) in the side channels of the Mekong delta in December 2003, April 2004 and October 2004. Dotted line indicates the linear regression. Error bars indicate standard deviation.




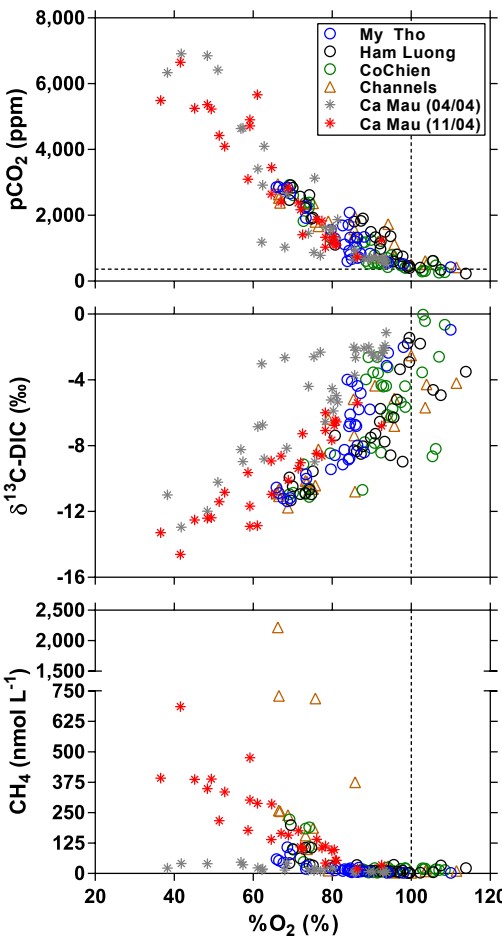

Fig. 9 - Distribution as a function oxygen saturation level (%$O_2$ in %) of the partial pressure of $CO_2$ (p$CO_2$ in ppm), stable isotope composition of dissolved inorganic carbon ($\delta^{13}$C-DIC in ‰), dissolved $CH_4$ concentration (nmol L$^{-1}$), in the three branches of the Mekong delta (My Tho, Ham Luong and Co Chien) and side channels, and in the mangrove creeks of the Ca Mau Province in April 2004 and October 2004. The vertical dotted line indicates $O_2$ saturation (100%), the horizontal line indicates the average atmospheric p$CO_2$ value.




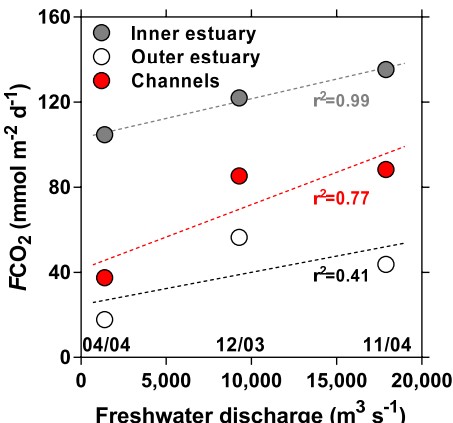

Fig. 10 – Average air-water $CO_2$ fluxes ($F$CO$_2$ in mmol m$^{-2}$ d$^{-1}$) in the inner and outer estuary and side channels of the Mekong delta as function of freshwater discharge (m$^3$ s$^{-1}$), in December 2003, April 2004 and October 2004. Sampling dates (MM/YY) are indicated in the bottom of the panel. Dotted lines indicate the linear regression lines.