# Peer review of "Carbon dynamics and CO2 and CH4 outgassing in the Mekong Delta"

_Biogeosciences, 2017_

## Referee Comment (RC1) · Anonymous Referee #1 · 28 Oct 2017

I read this manuscript with interest as it contains an excellent set of data in a large estuary. It is, however, a bit long especially the abstract. I have sampled in the Mekong Delta several times myself but the branch and place names are not familiar even to me. They should be omitted from the abstract.

My concerns are all minor:

1. The deltaTA/deltaDIC ratio of 0.55-0.87 is attributed only to aerobic degradation of organic matter and sulfate reduction. I don't understand why dissolution of calcite/aragonite and dolomite is not at play.

2. Many figures show good correlations but with only a few points. The high correlation coefficients, however, may not be statistically significant due to the small sample size. The p values should also be shown.

[Figure]

3. The authors stated, correctly, that CH4 has received much less attention on shelves compared to CO2. But, surely the authors know that Tseng et al., CSR, 2017,135,23-34 published CH4 data on shelves off the Mekong River delta.

4. A recent paper in JGR Biogeosciences(Huang et al., 2017, 122,1239) reported carbon export from many rivers around the SCS. A comparison would be of interest.

---

## Referee Comment (RC2) · Anonymous Referee #2 · 26 Dec 2017

The Ms by Borges et al sampled waters from the three branches in the Mekong delta and examined dissolved and particulate carbon and their C isotopes for carbon geochemical processes. The Ms provides important data for C characterization and controls in the important world River Mekong. Overall, the Ms is well organized with good writing style. I have some changes for improvement of the MS.

Abstract: This part seems to be longer than the journal guideline.

P2 L13: just say "tropical" is better due to that the study sites are located in the tropical climate biome.

P4 L 9: updated references should be added

P6 L16-17 two Ganges?

In the section of "2.1": The annual transports of sediment and solute by Mekong are

revised by Li and Bush (2015). I have noted the paper is cited by authors. Li, S.Y. †, Bush, R.T., 2015. Changing fluxes of carbon and other solutes from the Mekong River. Scientific Reports 5, 16005 DOI: 10.1038/srep16005

P9 What's the pore size?

P12 L3 Atmospheric CO2 of 362 ppm may be not a good data

P17 L15-18, how is the figure of 0.2 and 0.9 from?

P18 L12-13 Does the positive relation between $\Delta\delta$13C-DIC and %O2 can indicate organic matter degradation?

Table 1: Areal fluxes are presented as mean/median±S.D. will be better.

---

## Author Response (AR1)

**Chemical Oceanography Unit - MARE**
University of Liège
Institut de Physique (B5)
B-4000 Liège Belgium
**Alberto Borges**
**Senior research Associate FNRS**

Liège, 03 January 2018

Ji-Hyung Park
Guest-Editor Biogeosciences

Dear Dr. Park,

Please find enclosed the revised version of ms bg-2017-444.

We thank yours and the 3 referee's comments and suggestions for improvement.

We have reduced the sentence in the abstract you requested. As we replied to the reviewers, there is no size limit for the abstract in Biogeosciences and the length of our abstract reflects the descriptive nature of the paper of a rather large data-set.

We have defined partial pressure of CO2 and ppm. Text now reads: "The dissolved concentration of CO2 is expressed as pCO2 in parts per million (ppm), following Henry's law"

Please find hereafter, the replies to the 3 referees' comments.

We sincerely hope that the present version of the manuscript is acceptable for publication in Biogeosciences

Best regards
Alberto Borges

Tél. : +32-4-3663187 Fax : +32-4-3669729
E-mail : alberto.borges@ulg.ac.be
www.co2.ulg.ac.be

**Referee#1**

**Reviewer comment:** I read this manuscript with interest as it contains an excellent set of data in a large estuary. It is, however, a bit long especially the abstract. I have sampled in the Mekong Delta several times myself but the branch and place names are not familiar even to me. They should be omitted from the abstract.

**Reply:** We thank the reviewer for her/his positive evaluation of the paper. We kept the location names in the abstract; The Mekong delta is composed of different branches/rivers so it is necessary to name them in the abstract to clearly specify where we sampled; further south of our sampling area the Hau river (refer to figure 1) is an important component of the delta that was not sampled; in the paper we report two distinct data-sets, one of which is a mangrove site located in Ca Mau peninsula; this is a relatively extensive mangrove area that is frequently referenced by name in literature. In Biogeosciences, there is no size limit for the abstract. We acknowledge that our abstract is a bit long, but this reflects the descriptive nature of the paper of a rather large data-set.

**Reviewer comment:** 1. The deltaTA/deltaDIC ratio of 0.55-0.87 is attributed only to aerobic degradation of organic matter and sulfate reduction. I don't understand why dissolution of calcite/aragonite and dolomite is not at play.

**Reply:** We have added a discussion on the possibility of CaCO3 dissolution, and text now reads: "The slope of the linear regression of $\Delta$TA versus $\Delta$DIC ranged between 0.55 and 0.87. Such values might result from a combination of aerobic organic matter degradation (-0.2) and dissolution of $CaCO_3$ (or $CaMg(CO_3)_2$) (2.0). Accordingly, these values of relative changes of $\Delta$TA *versus* $\Delta$DIC would require that $CaCO_3$ dissolution corresponded to 34 and 48% of aerobic organic matter degradation, respectively. Such a large $CaCO_3$ dissolution is very unlikely in the Mekong delta because $Ca^{2+}$ and $Mg^{2+}$ showed conservative mixing as a function of salinity (Fig S3), and because particulate inorganic carbon (PIC) is relatively low in the Mekong delta compared to POC. The %PIC of TSM reported by Huang et al. (2017) is one order of magnitude lower (~0.1%) than the %POC of TSM we report (1-8%, Fig. 5)."

**Reviewer comment:** 2. Many figures show good correlations but with only a few points. The high correlation coefficients, however, may not be statistically significant due to the small sample size. The p values should also be shown.

**Reply:** We have added p values to the regressions

**Reviewer comment:** 3. The authors stated, correctly, that CH4 has received much less attention on shelves compared to CO2. But, surely the authors know that Tseng et al., CSR, 2017,135,23-34 published CH4 data on shelves off the Mekong River delta.

**Reply:** We now refer to the Tseng et al. (2017) CH4 data, text reads: "The $CH_4$ concentration in the most off-shore sampled station was indeed close to atmospheric equilibrium in April 2004 (2 nmol $L^{-1}$) for a salinity of 31.9, but was higher in October 2004 (17 nmol $L^{-1}$) reflecting the lower salinity of 17.0. These values encompassed the $CH_4$ concentrations of 4-6 nmol $L^{-1}$ reported by Tseng et al. (2017) 150 km away from the Mekong delta river mouth."

**Reviewer comment:** 4. A recent paper in JGR Biogeosciences(Huang et al., 2017, 122,1239) reported carbon export from many rivers around the SCS. A comparison would be of interest.

**Reply:** We have included the Huang et al. (2017) reference in the discussion of PIC (refer to above reply) and river TA and DOC concentrations, text now reads:

"Our TA values converge with the median (1082 µmol kg$^{-1}$) of a large data-set during 1972-1996 period from 42 stations in the lower Mekong delta compiled by the Mekong River Commission and reported by Li et al. (2014), and the average of TA data (1026 µmol kg$^{-1}$) acquired by Huang et al. (2017)."

and

"Within the freshwater zone (salinity <1), DOC values (2.4±0.2 mg L$^{-1}$, n=19) were within the range (0.9-5.1 mg L$^{-1}$) reported by Huang et al. (2017), and δ$^{13}$C-DOC values (-27.8 ± 0.6 ‰, n=19) were again consistent with a dominance of terrestrial C3 vegetation inputs, and close to values reported by Martin et al. (2013) slightly upstream in the lower Mekong"

**Referee#2**

**Reviewer comment:** The Ms by Borges et al sampled waters from the three branches in the Mekong delta and examined dissolved and particulate carbon and their C isotopes for carbon geochemical processes. The Ms provides important data for C characterization and controls in the important world River Mekong. Overall, the Ms is well organized with good writing style. I have some changes for improvement of the MS.

**Reply:** We thank the reviewer for her/his positive evaluation of the paper.

**Reviewer comment:** Abstract: This part seems to be longer than the journal guideline.

**Reply:** In Biogeosciences, there is no size limit for the abstract. We acknowledge that our abstract is a bit long, but this reflects the descriptive nature of the paper of a rather large data-set.

**Reviewer comment:** P2 L13: just say "tropical" is better due to that the study sites are located in the tropical climate biome.

**Reply:** We have kept "temperate" and "tropical", since the majority of $CO_2$ and $CH_4$ data have been primarily reported in temperate estuaries. Also, the convergence of data in the Mekong with data in temperate estuaries is informative.

**Reviewer comment:** P4 L 9: updated references should be added

Reply: We have added Testa et al. (2012) (doi: 10.1002/9781118412787.ch15)

**Reviewer comment:** P6 L16-17 two Ganges?

Reply: Typo was corrected.

**Reviewer comment:** In the section of "2.1": The annual transports of sediment and solute by Mekong are revised by Li and Bush (2015). I have noted the paper is cited by authors. Li, S.Y. y, Bush, R.T., 2015. Changing fluxes of carbon and other solutes from the Mekong River. Scientific Reports 5, 16005 DOI: 10.1038/srep16005

Reply: We have added the revised estimates of solute and solid transport given by Li and Bush (2015). Text now reads: "The annual sediment load was ~130-160 million tons in the 1960's and 110 million tons in the 1990's according to Milliman and Farnsworth (2011). Li and Bush (2015) report a less dramatic decrease of annual sediment load from 171 million tons for the pre-regulated period (1923-1991) to 168 million tons for the regulated period (1992-2007). Estimates of the annual solute transport ranges between 40 and 123 million tons (Meybeck and Carbonnel, 1975; Gaillardet et al., 1999; Li and Bush, 2015)"

**Reviewer comment:** P9 What's the pore size?

Reply: The porosity of GF/F filters is 0.7 μm. This information was added to text.

**Reviewer comment:** P12 L3 Atmospheric $CO_2$ of 362 ppm may be not a good data

Reply: Atmospheric pCO2 (mixing ratio in dry air) was around 376 ppm in 2003-2004. Once the atmospheric CO2 mixing ratio is converted from dry air to humidity saturated air (as required for CO2 flux computations), the values are around 362 ppm. We have added this information to text, that now reads: "The atmospheric $pCO_2$ values were converted from dry air to humidity saturated air using the water vapour formulation as function of salinity and temperature given by Weiss and Price (1980). For the three sampling periods, the dry air $CO_2$ mixing ratio averaged 376±4 ppm and the humidity saturated air $CO_2$ mixing ratio averaged 362±3 ppm."

**Reviewer comment:** P17 L15-18, how is the figure of 0.2 and 0.9 from?

Reply: We added this information, and text now reads : "The theoretical relative change of ΔTA *versus* ΔDIC was derived from the stoichiometry of biogeochemical reactions, based on Brewer and Goldman (1976) for aerobic respiration, on Smith and Key (1975) for $CaCO_3$ dissolution, and on Froelich et al. (1979) for anaerobic reactions"

**Reviewer comment:** P18 L12-13 Does the positive relation between __13C-DIC and %O2 can indicate organic matter degradation?

**Reply:** Yes, this relation is indeed consistent with expectations when organic matter degradation is the driving force behind the variations in both these parameters: organic matter degradation leads to O2 consumption and a preferential release of 12CO2 (since organic matter is isotopically light compared to the background DIC pool), leading to more negative delta 13C-DIC values

**Reviewer comment:** Table 1: Areal fluxes are presented as mean/median_S.D. will be better.

**Reply:** We have added to Table 1 the SD of the mean.

**Referee#3**

**Reviewer comment:** The manuscript Carbon dynamics in the Mekong Delta by Borges et al. present a valuable dataset of importance to understand the carbon dynamics not only of the Mekong River delta but also the understanding of this dynamics in large river estuaries. However, there are a few points that should be considered that would help to make the manuscript more suitable for publication.

**Reply:** We thank the reviewer for her/his positive evaluation of the paper.

**Reviewer comment:** The authors present data and discussion aiming to understand the carbon dynamics but information regarding, for example, the transport and degradation of carbon or factors controlling primary productivity is completely missing in the introduction. The manuscript introduction basically refers to CO2 and CH4 fluxes to the atmosphere, which is only a small part of the story presented. In section 2.1, despite the interesting and detailed information about geology and demography in upper basing, it is too long and most of this information is not essential for the manuscript. Instead more attention should be given to describe the carbon inputs from upstream the area, the vegetation, geomorphology, and hydrology of the delta and explaining where the delta starts. How the main channels differ from the side channels and mangrove creeks. Such information would be more useful to help interpret the results than social and economic information, which could be shortly mentioned in the introduction. Also, would be interesting to know how the tidal works in the area. How long time is each tidal cycle? What is the water level change in the mouth and in the most upstream site? Does the tide can significantly increase the residence time of the water?

**Reply:** The results and discussion of the paper are strongly focussed on $CO_2$ and $CH_4$ dynamics and their exchange with the atmosphere, and this is reflected in the introduction. The title of the ms was modified to put emphasis on $CO_2$ and $CH_4$. As part of the special issue on the "human impacts on carbon fluxes in Asian river systems", one of the aims of paper is to provide a baseline evaluation of C cycling in the Mekong delta that is faced with numerous threats. As such, we need to describe these threats and their consequences to the local population. Fundamental research on biogeochemical cycling in estuaries should be also made in a societal context, particularly in Asia. Finally, most of the information in the site description does relate to C cycling in the delta. For instance, in order to correctly interpret C cycling in the Mekong delta we need to take into account that nearly all of the original mangroves have been converted into shrimp farming ponds. We also included information on the riverine part of the Mekong River, and discuss in detail the carbon variations in the freshwater end-member of the delta.

Definition of the spatial extent of the delta and information on the tide has been added as requested by the reviewer, text now reads: "The upper limit of the delta (limit of the tidal influence) is the city of Phnom Penh in Cambodia, and at the coast it extends in the North from the mouth of the Saigon River to Cape Ca Mau in the South. The delta is meso-tidal with a mean tidal amplitude of 2.5 m at the estuarine mouth and a maximum tidal amplitude of 3.8 m" Information on nutrient inputs from the delta to the coastal zone has also been included, text now reads: "The nutrients inputs to the continental shelf from the Mekong delta sustain high phytoplankton growth in the Mekong river plume (Grosse et al., 2010) that is one the most productive areas of the South China Sea (Liu et al., 2002; Qiu et al., 2011; Gao et al., 2013; Loisel et al., 2017)"

**Reviewer comment:** In section 2.4 would be helpful for the reader if you explain the purpose of the mixing model informing what kind of information you can get from it and why this is important.

**Reply:** Text now reads: "Mixing models were used to investigate sources and sinks of TA, DIC, $O_2$ and $\delta^{13}C_{DIC}$ along the salinity gradient."

**Reviewer comment:** Despite the large list of papers regarding CO2 and CH4 fluxes from estuaries and large rivers none of them are mentioned in the discussion of section 3.4. Including this would improve the quality of the discussion.

**Reply:** We limited on purpose the comparison with large Asian estuaries (Pearl and Yangtze) that also border the China Sea in addition to the Mekong. Indeed, a general comparison with all World large estuarine systems would be a (very large) synthesis paper on itself, and such discussion is outside the scope of the present paper. We could have listed in a Table previous reported CO2 fluxes in all World large estuarine systems, although this is not very original on itself, nor very informative. If the reviewer has an interest in such comparisons, we invite her/him to consult the Cai et al. chapter (Carbon dioxide dynamics and fluxes in coastal waters influenced by river plumes, Chapter 7, pp. 155-173, Biogeochemical Dynamics at Large River-Coastal Interfaces: Linkages with Global Climate Change (Editors: T.S. Bianchi, M.A. Allison, and W.-J. Cai), 704 pp., Cambridge University Press).

**Reviewer comment:** Most of the discussion is about the controls of pCO2, which was already covered before and perhaps some information presented here would be more relevant to the earlier discussion. In this section, the authors should focus the parameters that would be more directly related to fluxes such as pCO2, wind and water velocity. pCO2 and wind speed are the two inputs for the fluxes calculation and despite pCO2 was nicely detailed, the wind speed data was not even mentioned and information regarding water velocity other than discharge would be interesting to be included. Sawakuchi et al 2017 (Front. Mar. Sci. 4:76), Alin et al 2011 and Borges et al 2004, shown that in large rivers and estuaries k is dependent on a mixture of pCO2, wind, and water flow and therefore is site specific. In tidal areas, water velocity could change significantly depending not only the season but the daily with the influence of the tides. This should also be covered here.

**Reply:** We thank the reviewer for reminding us of the importance of tidal currents in controlling the gas transfer velocity (k) in estuarine environments. As mentioned in the M&M we used the Raymond and Cole (2001) parameterisation as a function of wind speed. As also mentioned in the M&M, this parameterisation provides minimal (i.e conservative) estimates of k. We did not compute fluxes with other parameterisations because we did not have measurements of currents. Also it has been shown numerous times in literature that the use of the different parameterisations leads (unsurprisingly) to different values of fluxes, and we found superfluous to re-iterate this discussion in the present paper. Although we do not discuss in detail the variability of wind speed measurements, we have added the average wind speeds in Table 1.

**Reviewer comment:** To better show the spatial difference between main channels, side channels, mangrove creeks and the outer estuary. Statistical tests would be required to make the discussion more robust.

**Reply:** We have added p values of the regressions in figures as also required by Reviewer 1. We have added statistical tests where necessary.

**Reviewer comment:** Page 4, L21: April and Borges, 2004 is not in the References list.

**Reply:** Date was corrected from 2005 to 2004 in the reference list.

**Reviewer comment:** Page 5, L30-32: The same information is presented in the description of the river and should be removed from the introduction.

**Reply:** In the introduction are given the ranks compared to other World rivers. This information is important to situate the Mekong as a major World river. Text was not modified.

**Reviewer comment:** Page 6, L1-6: This is also in the methods section. Perhaps it would be better presenting it just there and remove it from the introduction.

**Reply:** Refer to above answer. Text was not modified.

**Reviewer comment:** Page 7, L27-29: This could be replaced by densely populated

**Reply:** Indeed, but we prefer to keep the quantitative information rather than replacing by a qualitative interpretation.

**Reviewer comment:** Page 9, L5-7: How much water was used and how it was collected and transferred to the serum bottles? Were there replicates? Consider the same for 13C-DIC.

**R**eply: Additional information was included and text now reads: "Water for the determination of $CH_4$ was sampled in duplicate with a silicone tube from the 1.7L Niskin bottle into 50 ml borosilicate serum bottles, allowing the flushing of 2-3 times the final volume, then poisoned with 100 µl of a saturated solution of $HgCl_2$ sealed with a butyl stopper and crimped with an aluminium cap"

**Reviewer comment:** Page 10, L22-24: Recent papers have shown that estimated k values for CO2 are different from CH4. Thus, a single model would not be the best way to tackle the flux estimates. If not possible to use different models for each gas, at least this limitation should be mentioned.

**Reply:** We are familiar with the recent papers dealing with differences in k for CO2 and CH4. However, this has solely shown with floating chamber measurements, an approach prone to several methodological caveats. Further, there is little understanding of the actual mechanisms that could explain such differences of k among gases. Anyway, we believe this level of refinement in the computations of $CO_2$ and $CH_4$ values is superfluous given the enormous spatial and seasonal variations of the fluxes driven by those of the concentrations.
Our data we will not help resolving this problem; using a single but broadly used k600-wind relationship for our flux calculations has the advantage to allow simple corrections if a new parameterization is published for the Mekong delta in the future

**Reviewer comment:** Page 10, L24-28: Please add here the atmospheric CO2 retrieved from NOAA and the distance from the station to the mouth of the Mekong River.

**Reply:** Text now reads: "During the three sampling periods, the dry air $CO_2$ mixing ratio averaged 376±4 ppm and the humidity saturated air $CO_2$ mixing ratio averaged 362±3 ppm."

**Reviewer comment:** Page 12, L11-13: Please show the correlation between pCO2 and O2.

**Reply:** We have added a figure showing the highly significant correlation between pCO2 versus %O2 and $\delta^{13}$C-DIC. Text now reads: "The $pCO_2$ values in freshwaters were significantly correlated to %$O_2$ (Fig. 4) indicating biological control of both these variables. Similarly, the correlation between $pCO_2$ and $\delta^{13}$C-DIC (Fig. 4) results from the degradation of organic matter that leads to a preferential release of $^{12}CO_2$ (since organic matter is isotopically light compared to the background DIC pool), leading to more negative delta $\delta^{13}$C-DIC values."

**Reviewer comment:** Page 12, L21-23. Please present the 13C-DIC values here.

**Reply:** Text now reads: "The impact of biological activity on $CO_2$ dynamics in the uppermost freshwater part of the estuary, was confirmed by $\delta^{13}$C-DIC values that were higher in April 2004 (-8.7±0.4 ‰, n=5) compared to December 2003 (-10.6±0.6, n=6 ‰) and October 2004 (-10.9±0.3 ‰, n=15)"

**Reviewer comment:** Page 12, L23-25: What happens with other variables like POC, DOC, and TSM? The same discharge can be observed during the rising and falling water period, however, there will be different inputs of these materials depending if it is rising or falling waters. Furthermore, please consider adding in Fig 4 the channels and the outer estuary, similarly as you did in Figure 10.

**Reply:** As mentioned above, the primary focus of the paper is CO2 and CH4 dynamics, so we plotted these data plus those that are immediately explicative such as O2 and $\delta^{13}$C-DIC. Plotting other variables POC, etc… would bring little added value given the limited number of data (only 3 three cruises, when change of POC, TSM as a function of discharge are notoriously variable). We did not add the data in channels and outer estuary as a function of discharge, as they are also affected by local mixing processes and production/removal processes (inputs from shrimp farms) that are independent of discharge.

**Reviewer comment:** Page 13, L4-5: Are these other tropical estuaries large systems, and where they are located?

**Reply:** We added in text that the cited estuaries are located in Kenya. These are smaller systems than the Mekong, although we did not find relevant to mention this in text.

**Reviewer comment:** Page 13, L5-7: Neither the O2 decrease or the pCO2 increasing patterns are not very clear in Fig 3. In April 2004 pCO2 increased from km 40 to 60, but then it decreased toward the mouth.

**Reply:** Indeed, that is mentioned P12 L30.

**Reviewer comment:** Page 13, L11: Please inform if the Phnom Penh is similar in terms of size/discharge to the Mekong

**Reply:** Phnom Penh is a city located on the Mekong river.

**Reviewer comment:** Page 13, L16-17: Considering this finding you could exclude the long-term change explanation given above.

**Reply:** The sequence of sentences provides a reasoning with hypothesis and test of hypothesis. We did not change text.

**Reviewer comment:** Page 13, L22-30: You need to make a link to how all this information regarding these cations affects DIC and consequently pCO2. The way it is presented it seems out of context and does not contribute to the understanding of the C dynamics.

**Reply:** We assume that readers will be aware that total alkalinity in freshwaters mainly corresponds to $HCO_3^-$, and that DIC is predominately constituted by $HCO_3^-$. Rather than adding a substantial amount of text to explain this, we opted not to change the text.

**Reviewer comment:** Page 14, L6: The Oct 2004 cruise you have samples collected farther out the mouth in the plume. This could be pure or almost pure ocean water and may be related to this very low pCO2 values observed. In addition later in Page 15, L5 you have mentioned that there was a phytoplankton bloom in the outer estuary.

**Reply:** Yes, we went more offshore due to very good weather conditions during this cruise, but the highest salinity in Oct 2004 was 27 lower than the highest salinity in April 2004 of 31. Ocean ("pure marine") salinity is typically 35, so well above the observed max salinities in the river plumes. Additionally, the phytoplankton growth is observed in the salinity gradient from 10 to 27 and not only in the high salinity range. Finally, a higher primary production in October than during the other two cruises is consistent with remote sensed seasonal cycles of phytoplankton biomass. Text now reads "Reported seasonal cycles of remote sensed Chlorophyll-*a* concentration also indicate higher phytoplankton biomass and primary production in October compared to April and December (Gao et al. 2013; Loisel et al. 2017)."

**Reviewer comment:** Page 14, L14: Could the depleted 13 C signature in the DIC be related with a large input of C4 plant material?

**Reply:** The DOC and POC stable isotope composition indicates a large predominance of terrestrial C3 material in the freshwaters of the Mekong. While there are some sugar cane (i.e. C4 vegetation) plantations in the area, their contribution is likely modest and cannot result in massive inputs that could impose the signature of delta 13C-DIC, without altering the isotopic signal and concentrations of aquatic DOC and POC. Hence, the explanation we give (mix of degradation organic matter of C3 origin and rock dissolution) is most likely.

**Reviewer comment:** Page 14, L17-18: What would be the expected 13C signature in the DIC from the weathering carbonate and silicate?

**Reply:** The delta 13C-DIC from dissolution of CaCO3 comes ½ from CaCO3 (delta 13C =

0) and ½ from CO2 used in dissolution (typically from organic matter degradation). delta 13C-DIC from dissolution of silicate rocks comes exclusively from CO2 used in dissolution (typically from organic matter degradation). This is relatively trivial information, we did not change text.

**Reviewer comment:** Page 15, L7: On page 14 L6 you say that the lowest pCO2 was 314. Which is the right value?

**Reply:** One value is the absolute minimum, the other is the value in the most marine station (highest salinity). We did not change text.

**Reviewer comment:** Page 15, L15-24: Seidel et al 2015 (Marine Chemistry 177,p 218–231) and Medeiros et al 2015. (Global Biogeochem. Cycles, 29, p677–690) may give you valuable information regarding this discussion.

**Reply:** We thank the reviewer for the suggested references, but we considered a comparison with the Amazon to be outside the scope of this already very long paper.

**Reviewer comment:** Page 15, L28-30: Despite the trends observed in Fig 5 for the variables, these different patterns does not seem to be statistically significant. Please add the proper statistical tests results to make your point more convincing.

**Reply:** We added p values.

**Reviewer comment:** Page 16, L26-29: You have mentioned above that %POC, TSM and 13C POC in the side channels indicate large primary production. I am not sure if I got it right, but the model seems to not see the same trend, is that right?

**Reply:** Both are occurring. Phytoplankton growth affects $\delta^{13}$C-POC, and heterotrophy leads to high $CO_2$, $CH_4$, and low $O_2$. We have added a sentence to clarify this, and text now reads: "Although there is indication of phytoplankton development based on $\delta^{13}$C-POC (see above), the overall system was net heterotrophic leading to accumulation of $CO_2$, $CH_4$ and light DIC, and decrease of $O_2$."

**Reviewer comment:** Page 17, L10-13: If the side channels have lower water velocity than the main channels, higher sediment deposition is expected in this areas differently from the main channels were the water flow would wash out the fine organic sediment. If you have such condition then these shallow side channels could accumulate C fueling anaerobic degradation and CH4 production.

**Reply:** We fully agree, and that's indeed what is probably going on, but we do not have data on currents to demonstrate that, so we did not include this in text.

**Reviewer comment:** Figure 1. Add spatial scale bar and a North arrow.

**Reply:** We added the scale bar. We did not add a "North arrow", the plots are oriented to the North, as by convention.

**Reviewer comment:** Figure 3. Please consider changing the colored circles into different shapes. This would make easier to distinguish between rivers.

**Reply:** We kept the original symbols that guide the reader in interpreting the plots by discriminating the side channels (triangles) from the main estuarine branches (all with circles).

**Reviewer comment:** Figure 4. Would be interesting to see the channels and the outer estuary here as well, similarly to Fig 10.

**Reply:** Please refer to above reply on similar comment.

**Reviewer comment:** Figures 5, 6, 7 and 9. Same as figure 3.

**Reply:** Please refer to above reply on similar comment.

[revised manuscript text omitted]